# The tumor suppressor PTPRK promotes ZNRF3 internalization and is required for Wnt inhibition in the Spemann organizer

**Ling-Shih Chang[1†], Minseong Kim[1†], Andrey Glinka[1], Carmen Reinhard[1], Christof Niehrs[1,2]\***

[1]Division of Molecular Embryology, DKFZ-ZMBH Alliance, Deutsches Krebsforschungszentrum (DKFZ), Heidelberg, Germany; [2]Institute of Molecular Biology (IMB), Mainz, Germany

**Abstract** A hallmark of Spemann organizer function is its expression of Wnt antagonists that regulate axial embryonic patterning. Here we identify the tumor suppressor Protein tyrosine phosphatase receptor-type kappa (PTPRK), as a Wnt inhibitor in human cancer cells and in the Spemann organizer of *Xenopus* embryos. We show that PTPRK acts via the transmembrane E3 ubiquitin ligase ZNRF3, a negative regulator of Wnt signaling promoting Wnt receptor degradation, which is also expressed in the organizer. Deficiency of *Xenopus* Ptprk increases Wnt signaling, leading to reduced expression of Spemann organizer effector genes and inducing head and axial defects. We identify a '4Y' endocytic signal in ZNRF3, which PTPRK maintains unphosphorylated to promote Wnt receptor depletion. Our discovery of PTPRK as a negative regulator of Wnt receptor turnover provides a rationale for its tumor suppressive function and reveals that in PTPRK-RSPO3 recurrent cancer fusions both fusion partners, in fact, encode ZNRF3 regulators.

**\*For correspondence:**
niehrs@dkfz-heidelberg.de

[†]These authors contributed equally to this work

**Competing interests:** The authors declare that no competing interests exist.

## Introduction

The Spemann organizer is an evolutionary conserved signaling center in early vertebrate embryos, which coordinates pattern formation along the anterior–posterior, dorsal–ventral, and left–right body axes (*Harland and Gerhart, 1997*; *De Robertis et al., 2000*; *Niehrs, 2004*). In amphibian embryos, the organizer corresponds to the upper dorsal blastopore lip, constituting mostly dorsal mesendoderm. Molecularly, the Spemann organizer functions by negative regulation of BMP, Nodal, and Wnt signaling. Wnt/β-catenin signaling plays a key role in antero-posterior (a-p) patterning the *Xenopus* neural plate where a signaling gradient promotes posterior fate (*Hoppler et al., 1996*; *Hoppler and Moon, 1998*; *Kiecker and Niehrs, 2001*), a role, which is evolutionary conserved (*Niehrs, 2010*). Various Wnt antagonists or membrane-bound Wnt inhibitors are expressed in neural-inducing dorsal mesoderm and/or the prospective neuroectoderm itself to promote organizer function, and to pattern the neural plate, including *cerberus, frzb, dkk1, shisa, tiki, notum, angptl4*, and *bighead* (*Bouwmeester et al., 1996*; *Leyns et al., 1997*; *Glinka et al., 1998*; *Yamamoto et al., 2005*; *Zhang et al., 2012*; *Cruciat and Niehrs, 2013*; *Zhang et al., 2015*; *Kirsch et al., 2017*; *Ding et al., 2018*). Thus, the *Xenopus* Spemann organizer has been a treasure trove for the discovery of negative Wnt regulators, informing on their function in cell and tissue homeostasis as well as in disease (*Cruciat and Niehrs, 2013*). With regard to the latter, activation of Wnt/β-catenin signaling is a ubiquitous feature in colorectal cancer (*Nusse and Clevers, 2017*; *Zhan et al., 2017*) and thus comprehensive understanding of Wnt regulators is a key towards developing therapeutic approaches for cancer.

**eLife digest** How human and other animals form distinct head- and tail-ends as embryos is a fundamental question in biology. The fertilized eggs of the African clawed frog (also known as *Xenopus*) become embryos and grow into tadpoles within two days. This rapid growth makes *Xenopus* particularly suitable as a model to study how animals with backbones form their body plans.

In *Xenopus* embryos, a small group of cells known as the Spemann organizer plays a pivotal role in forming the body plan. It produces several enzymes known as Wnt inhibitors that repress a signal pathway known as Wnt signaling to determine the head- and tail-ends of the embryo.

Chang, Kim et al. searched for new Wnt inhibitors in the Spemann organizer of *Xenopus* embryos. The experiments revealed that the Spemann organizer produced an enzyme known as PTPRK that was essential to permit the head-to-tail patterning of the brain. PTPRK inhibited Wnt signaling by activating another enzyme known as ZNRF3.

Previous studies have shown that defects in Wnt signaling and in the activities of PTPRK and ZNRF3 are involved in colon cancer in mammals. Thus, these findings may help to develop new approaches for treating cancer in the future.

Wnt/β-catenin signaling operates via the transcriptional coactivator β-catenin, whose level is tightly regulated by Axin/APC/GSK3 destruction complex-mediated phosphorylation, ubiquitination, and proteasomal degradation. Binding of Wnt ligands to Frizzleds (FZDs) receptors and co-receptors of the LDL Receptor Related Protein (LRP) −5 and −6 family inhibits GSK3 and the destruction complex, hence β-catenin can accumulate and translocate to the nucleus (*Nusse and Clevers, 2017*; *Zhan et al., 2017*). In addition, Wnt signaling is also elaborately tuned at the receptor level (*Niehrs, 2012*; *Kim et al., 2013*; *Green et al., 2014*). For example, the single transmembrane E3 ligases ZNRF3/RNF43 ubiquitylate and downregulate FZDs and LRP6, imposing negative feedback control on Wnt signaling. R-spondin ligands sequester ZNRF3/RNF43 with LGR4/5/6 and lead to the membrane clearance of ZNRF3/RNF43 (*Carmon et al., 2011*; *de Lau et al., 2011*; *Glinka et al., 2011*; *Hao et al., 2012*; *Koo et al., 2012*). Thereby, R-spondins increase the membrane abundance of Wnt receptors and potentiate Wnt signaling.

Aberrant Wnt/R-spondin/ZNRF3 signaling is implicated in tumorigenesis, where 7% of colon cancer and 31% of serrated adenoma samples harbor *RSPO3* gene fusions with the neighboring *Protein tyrosine phosphatase receptor-type kappa* (*PTPRK*) gene (*Seshagiri et al., 2012*; *Sekine et al., 2016*). In these gene fusions, the signal sequence of PTPRK is fused to RSPO3, reducing PTPRK and leading to elevated RSPO3 protein levels, which in transgenic mouse models are sufficient to drive tumor initiation (*Han et al., 2017*).

The tumor-promoting effect of *PTPRK-RSPO3* gene fusions is solely attributed to elevated R-spondin levels, while little attention has been paid to a possible role of *PTPRK* in this context. PTPRK belongs to R2B subfamily of Receptor type protein tyrosine phosphatases (RPTP) (*Jiang et al., 1993*), which contain an adhesion molecule-like extracellular domain and a cytoplasmic tyrosine phosphatase domain (*Lee et al., 2015*). PTPRK can be cleaved by multiple proteases to generate a soluble intracellular fragment that can translocate into the nucleus (*Anders et al., 2006*; *Tonks, 2006*). Hence, PTPRK can have a variety of substrates from the membrane to the nucleus. PTPRK dephosphorylates and inactivates oncogenic proteins such as STAT3, EGFR and CD133, is frequently downregulated in human cancers, and is considered a tumor suppressor (*McArdle et al., 2001*; *Flavell et al., 2008*; *Tarcic et al., 2009*; *Assem et al., 2012*; *Scrima et al., 2012*; *Mo et al., 2013*; *Sun et al., 2013*; *Chen et al., 2015*; *Shimozato et al., 2015*).

Here, we report that not only *RSPO3* but also its fusion partner *PTPRK* encodes a regulator of ZNRF3 and Wnt/β-catenin signaling. In *Xenopus* embryos, both *ptprk* and *znrf3* are expressed in the Spemann organizer and are required to inhibit Wnt signaling to promote early embryonic axial patterning and head formation. PTPRK binds to ZNRF3, causes its tyrosine-dephosphorylation at a conserved '4Y' internalization signal, and enhances ZNRF3-mediated Wnt receptor turnover. Thus, PTPRK has the opposite function of RSPO3, promoting- instead of preventing Wnt receptor removal.

Our study suggests that in *PTPRK-RSPO3* gene fusions, truncation of PTPRK and increased expression of RSPO3 in fact work in the same direction, impairing ZNRF3 to augment Wnt signaling.

## Results

### PTPRK is a negative regulator of Wnt/β-catenin signaling

To uncover novel regulators of Wnt/β-catenin signaling, a genome-wide small interfering RNA (siRNA) screen using Wnt reporter assay as a readout was previously performed (*Cruciat et al., 2010*) and PTPRK was discovered as a potential candidate. In the H1703 human lung adenocarcinoma cell line, knockdown of PTPRK enhanced Wnt3a induced signaling in Topflash reporter assays (*Figure 1A* and *Figure 1—figure supplement 1A*) as well as expression of the endogenous Wnt target gene *AXIN2* (*Figure 1B* and *Figure 1—figure supplement 1B*). si*PTPRK* also increased cytosolic β-catenin levels and nuclear accumulation of β-catenin upon Wnt3a treatment (*Figure 1C–D*). PTPRK was reported to promote membrane association of β-catenin (*Novellino et al., 2008*), but we found no change in β-catenin in the membrane fraction in si*PTPRK* treated cells (*Figure 1—figure*

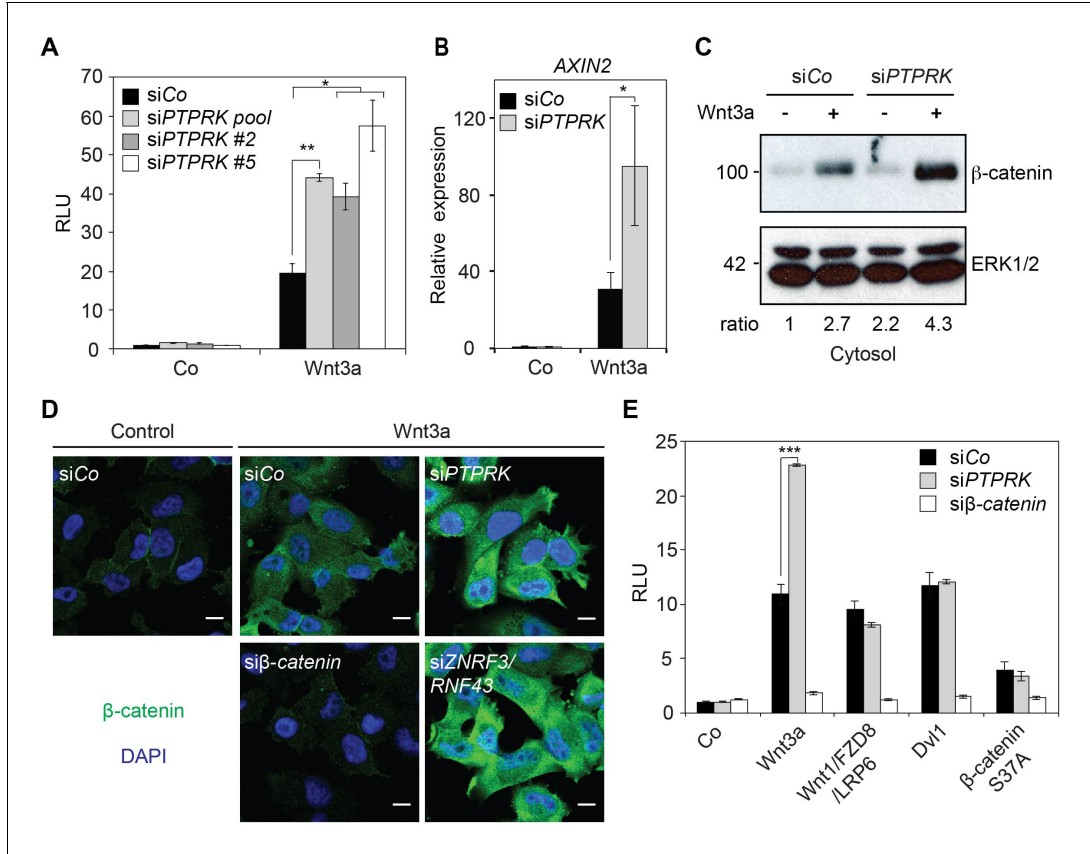

**Figure 1.** PTPRK inhibits Wnt/β-catenin signaling at the receptor level. (**A**) Topflash reporter assay in H1703 cells upon si*Co*, si*PTPRK* pool, or single si*PTPRK*s (si*PTPRK* #2, #5) transfection, with or without overnight Wnt3a treatment. Further experiments were done with si*PTPRK* #2. (**B**) qRT-PCR analysis of *AXIN2* in H1703 cells treated with Wnt3a overnight upon si*Co*, si*PTPRK* transfection. (**C**) Western blot analysis of cytosolic β-catenin in H1703 cells upon si*Co* or si*PTPRK* transfection. Cells were treated with Wnt3a for 2 h before harvest and permeabilized with 0.05% Saponin. Ratio, relative levels of β-catenin normalized to ERK1/2. (**D**) Immunofluorescence microscopy showing nuclear and cytosolic β-catenin in H1703 cells. Cells were transfected with the indicated siRNAs and treated with Wnt3a for 2 h. si*β-catenin* and si*ZNRF3*/si*RNF43* were used as negative and positive control, respectively. (**E**) Topflash reporter assay in H1703 cells upon PTPRK or β-catenin knockdown. Topflash activity was stimulated by overnight treatment of Wnt3a, or transfection of Wnt1/Fzd8/LRP6, Dvl1, or hβ-catenin S37A. Data in all graphs are displayed as means ± SD, and show one representative of multiple independent experiments with three biological replicates. RLU, relative light units. *p<0.05 **p<0.01, ***p<0.001.

The online version of this article includes the following figure supplement(s) for figure 1:

**Figure supplement 1.** Knockdown effects of si*PTPRK*s.

*supplement 1C*). Furthermore, in epistasis experiments, si*PTPRK* increased Topflash reporter activity only when the Wnt reporter was activated by Wnt3a but not following transfection of Wnt1/Fzd8/LRP6, Dvl1 (Dishevelled 1), or constitutively active β-catenin (S37A) (*Figure 1E*). PTPRK affected Wnt signaling only upon knockdown, but not overexpression (*Figure 1—figure supplement 1D*). Moreover, unlike other negative Wnt regulators such as Naked, APC, or GSK3, which act universally, Wnt inhibition by PTPRK was not observed in e.g. HEK293T cells (*Figure 1—figure supplement 1E*), and hence PTPRK seems to act cell-type specifically. In addition, when we tested other RPTPs expressed in H1703 cells (based on available RNAseq databases), si*PTPRK* showed the strongest effect on inducing *AXIN2* expression, besides si*PTPRF* (*Figure 1—figure supplement 1F*). Taken together, these results indicate that PTPRK acts at the receptor level to inhibit Wnt/β-catenin signaling in H1703 cells.

## *Ptprk* is expressed in the Spemann organizer and is required to inhibit Wnt signaling

We next studied the role of PTPRK in vivo in the African clawed frog *Xenopus tropicalis*, since the role of early Wnt signaling in the Spemann organizer of amphibian embryos is well-established (*Niehrs, 2004*). Analysis of *Xenopus ptprk* by qRT-PCR showed that it was expressed maternally and continued to be expressed at similar levels during gastrulation, increasing with organogenesis (*Figure 2—figure supplement 1A*). By whole-mount in situ hybridization, *ptprk* was expressed in the animal hemisphere of blastula embryos (*Figure 2—figure supplement 1B*). In early gastrulae, *ptprk* was prominently expressed in the Spemann organizer (*Figure 2A*). While clearly enriched on the dorsal side, *ptprk* expression was not exclusive to the organizer but was also weakly detected in ventral cells. Interrogating a database derived from RNAseq of *Xenopus* genes with ranked organizer-specific expression (*Ding et al., 2017*) confirmed differential expression of *ptprk* on the dorsal side, but with lower enrichment than some other 'organizer genes' (*Figure 2D*). In neurulae and tailbud embryos, *ptprk* was most prominently expressed in the notochord (*Figure 2B–C*), an organizer derivative, which plays a critical role in neural patterning (*Hemmati-Brivanlou et al., 1990*; *Yamada et al., 1991*; *Roelink et al., 1994*; *Barnett et al., 1998*; *Wilson and Maden, 2005*). Low expression was detected in the neural plate, as well as branchial arches and dorsal lateral plate (*Figure 2B*). We conclude that *Xenopus ptprk* is prominently expressed in the Spemann organizer and notochord.

We depleted Ptprk by Morpholino antisense oligo (Mo) injection, targeting the splice site between exon 1 and intron 1 of *Xenopus tropicalis ptprk*, and efficiently reduced *ptprk* mRNA (*Figure 2—figure supplement 1C*). Microinjection of *ptprk* Mo in *Xenopus* ('morphants') led to reduced head structures and shortened body axis, which was rescued by coinjection of untargeted human *PTPRK* mRNA (*Figure 2E* and *Figure 2—figure supplement 1D*), demonstrating Mo specificity. To further confirm specificity of these defects, we carried out CRISPR/Cas9 mediated *ptprk* gene editing. A single guide RNA (sgRNA) was designed to target a sequence within *ptprk* exon one and the genome modification was confirmed by StuI enzyme digestion (*Figure 2—figure supplement 1F–G*). The *ptprk* genome-edited embryos ('crispants') showed the same phenotype as *ptprk* morphants. Anterior and tail formation defects are characteristically observed following overactivation of zygotic Wnt signaling (*Christian and Moon, 1993*) and expectedly *Wnt8* DNA overexpression phenocopied the *ptprk* morphant and crispant phenotype (*Figure 2E* and *Figure 2—figure supplement 1D–E*). Concordantly, depletion of Ptprk upregulated Wnt-induced Topflash activity in *Xenopus* embryos, both in morphants (*Figure 2F*) and crispants (*Figure 2G*). Increased Wnt activity in *ptprk* morphants was restored by human wild-type *PTPRK* RNA but not by an intracellular domain deletion mutant (*PTPRK-ΔC*) or phosphatase-dead mutants (*PTPRK-CS, PTPRK-DA*) (*Figure 2F,H–I*), indicating that the tyrosine phosphatase activity is essential for Wnt inhibition. The importance of PTPRK phosphatase activity in Wnt regulation was also confirmed in H1703 cells (*Figure 2—figure supplement 1H*). We conclude that *ptprk* depletion upregulates Wnt signaling and phenocopies Wnt overactivation during early *Xenopus tropicalis* development, supporting that Ptprk is a negative regulator of Wnt signaling not only in H1703 cancer cells but also in vivo.

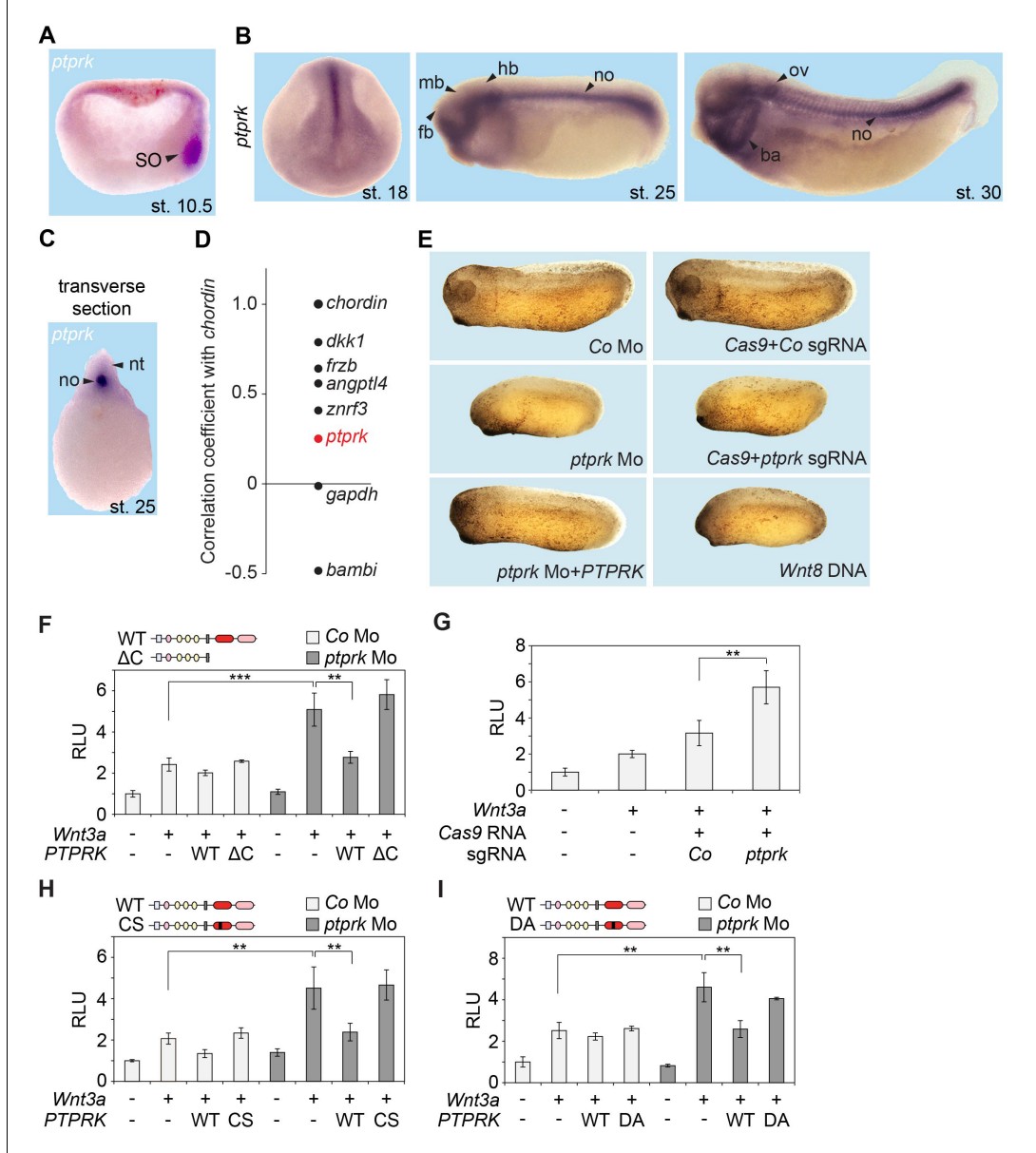

**Figure 2.** Ptprk inhibits Wnt signaling in the *Xenopus* Spemann organizer. (**A–C**) In situ hybridization of *ptprk* in *Xenopus tropicalis* at (**A**) gastrula (hemisected, dorsal to the right), (**B**) neurula, tailbud, and tadpole stages, and in (**C**) transverse dissected tailbud embryo. ba, branchial arches; fb, forebrain; hb, hindbrain; mb, midbrain; no, notochord; nt, neural tube; ov, otic vesicle; SO, Spemann organizer. (**D**) Data mining using data from ***Ding et al. (2017)***, showing gene expression correlation with a dorsal/organizer marker *chordin*. *Xenopus dkk1*, *frzb*, and *angptl4* are known organizer-expressed genes, *gapdh* is shown as housekeeping gene, and *bambi* is a ventrally expressed gene. (**E**) Representative phenotypes of tailbud stage *Xenopus tropicalis* embryos injected animally at 2- to 8 cell stage and as indicated. For quantification, see ***Figure 2—figure supplement 1D–E***. (**F–I**) Topflash reporter assays performed with neurulae (stage 18). Embryos were injected animally at 2- to 8 cell stage (**F, H–I**) or one cell stage (**G**) with reporter plasmids and the indicated mRNAs and Mos. Domain structures of WT PTPRK and mutants are shown on top. Normalized Topflash activity of Co-injected embryos only with reporter plasmids was set to 1. Data in all graphs are displayed as means ± SD, and show one representative of multiple independent experiments with three biological replicates. RLU, relative light units. \*\*p<0.01, \*\*\*p<0.001.

The online version of this article includes the following figure supplement(s) for figure 2:

**Figure supplement 1.** Spatiotemporal expression and knockdown of *ptprk* in *Xenopus* embryos.

## Ptprk promotes Spemann organizer function

Inhibition of zygotic Wnt signaling is required for normal organizer gene expression (*Hoppler et al., 1996*; *Kirsch et al., 2017*; *Ding et al., 2018*). Consistently, microinjection of *ptprk* Mo downregulated expression of Spemann organizer effector genes, including *chordin* (*chd*), *goosecoid* (*gsc*) and *Xnot2* (*Figure 3A* and *Figure 3—figure supplement 1A–C*). Zygotic Wnt signaling inhibits anterior neural gene expression, which is counteracted by Wnt antagonists. To corroborate the role of Ptprk in Wnt-mediated anterior neural patterning, we analyzed expression of the forebrain markers, *bf1* and *otx2*. Unilateral injection of *ptprk* Mo with lineage tracer downregulated *bf1* and *otx2*

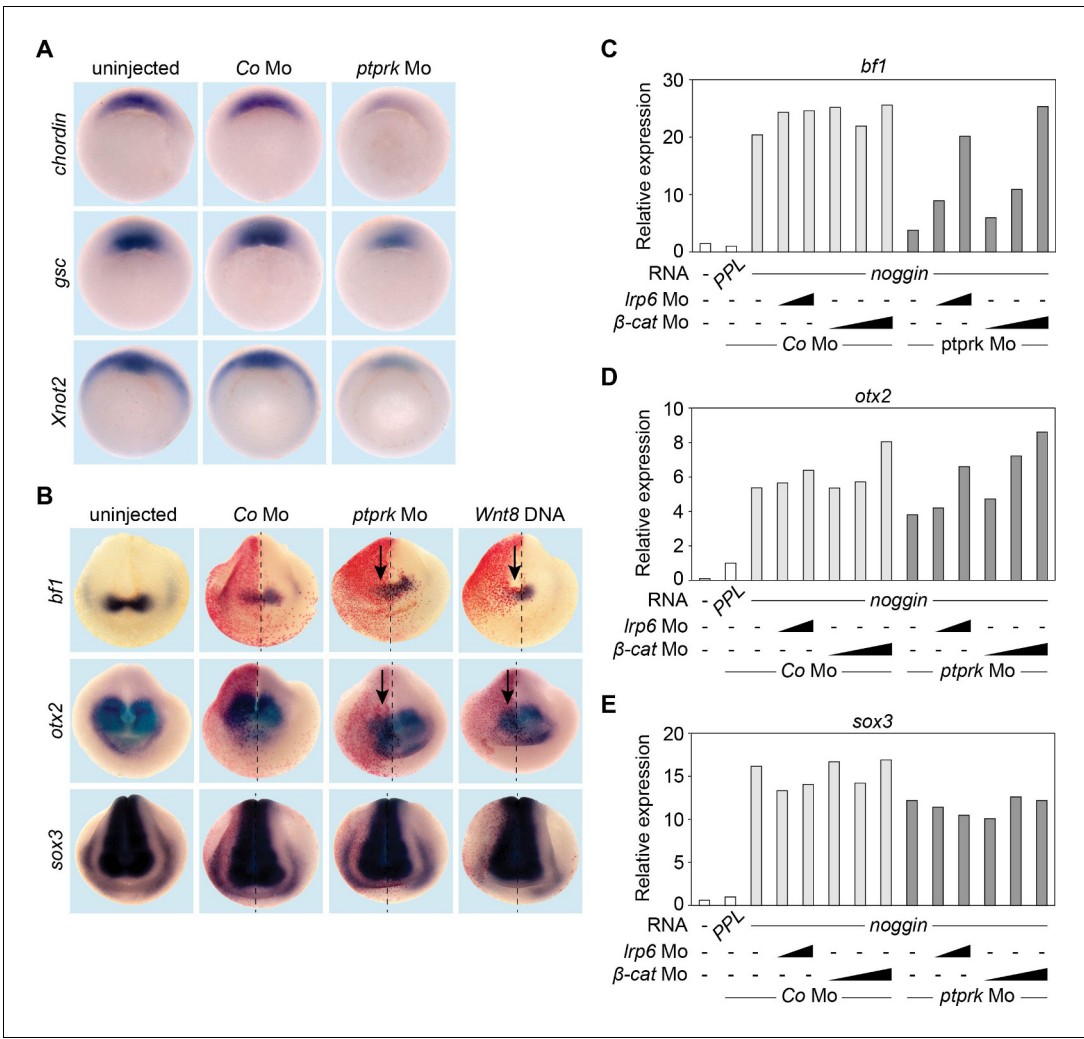

**Figure 3.** Ptprk regulates Spemann organizer function by inhibiting Wnt signaling. (**A**) Whole mount in situ hybridization of *chordin*, *gsc*, and *Xnot2* in gastrula embryos (stage 10.5). Embryos were injected at 2- to 8 cell stage animally with *Co* or *ptprk* Mo. For quantification, see *Figure 3—figure supplement 1A–C*. (**B**) Whole mount in situ hybridization of forebrain marker *bf1* and *otx2*, and pan-neural marker *sox3* in neurula embryos (stage 18). Embryos were injected at 4- to 8 cell stage unilaterally in animal blastomeres as indicated (β-galactosidase lineage tracer in red; arrows mark injected side). For quantification, see *Figure 3—figure supplement 1D–F*. (**C–E**) qRT-PCR analysis showing the expression of (**C**) *bf1*, (**D**) *otx2* and (**E**) *sox3* in *Xenopus tropicalis* animal cap explants. Embryos were injected animally at 2- to 8 cell stage as indicated. *Xenopus noggin* mRNA was injected to induce to neural fate in animal cap explants. Animal caps were excised at stage 9 and harvested at stage 18. The expression of each gene was normalized to *odc*. *PPL* and *Co* Mo injected embryos were set to 1. Data show one representative experiment of at least three independent experiments with similar results.

The online version of this article includes the following figure supplement(s) for figure 3:

**Figure supplement 1.** Ptprk regulates Spemann organizer function.

expression on the injected side, as did *Wnt8* DNA overexpression (*Figure 3B* and *Figure 3—figure supplement 1D–E*). Neural induction was not impaired as expression of the pan-neuronal marker *sox3* was unaffected (*Figure 3B* and *Figure 3—figure supplement 1F*). We carried out rescue experiments in *Xenopus noggin*-neuralized animal cap explants (*Lamb et al., 1993*). BMP4 inhibition by *noggin* mRNA injection expectedly induced neural markers, and *ptprk* Mo reduced the expression of *bf1* and *otx2*, but not *sox3* (*Figure 3C–E* and *Figure 3—figure supplement 1G*). Importantly, knockdown of *lrp6* or *β-catenin* using established Mos (*Heasman et al., 2000*; *Hassler et al., 2007*) rescued the effects of *ptprk* Mo on *bf1* or *otx2* expression in a dose-dependent manner (*Figure 3C–D*). These results confirm that the reduction of forebrain markers in *ptprk* morphants resulted from increased Wnt activity. We conclude that Ptprk promotes Spemann organizer function by negatively modulating Wnt/β-catenin signaling at the Lrp6 receptor level in vivo.

## PTPRK regulates surface levels of Wnt receptors through ZNRF3

The in vitro and in vivo data clearly indicated that PTPRK regulates Wnt signaling at the receptor level. Moreover, PTPRK depletion increased not only LRP6 phosphorylation/activation, but also total LRP6 levels in H1703 cells and *Xenopus* embryos (*Figure 4A–C* and *Figure 4—figure supplement 1A,C*), without affecting *LRP6* mRNA levels (*Figure 4—figure supplement 1B,D*). This suggests that PTPRK directly or indirectly reduces LRP6 protein levels. The transmembrane E3 ligases ZNRF3 and its homolog RNF43 are key negative regulators of Wnt receptor levels at the plasma membrane (*Hao et al., 2012*; *Koo et al., 2012*). Hence, we explored if PTPRK may act through ZNRF3/RNF43.

PTPRK depletion upregulated LRP6 levels similarly to knockdown of ZNRF3 and RNF43 in H1703 cells (*Figure 4B* and *Figure 4—figure supplement 1A*) as well as in *Xenopus* embryos (*Figure 4C* and *Figure 4—figure supplement 1C*). ZNRF3/RNF43 degrade not only LRP6 but also FZD receptors (*Hao et al., 2012*; *Koo et al., 2012*). We therefore monitored FZD levels at the plasma membrane by flow cytometry using a pan-FZD antibody (OMP-18R5) (*Gurney et al., 2012*; *Hao et al., 2012*). Consistently, not only si*ZNRF3/RNF43* but also si*PTPRK* increased FZD cell surface levels (*Figure 4D*). Examining their epistasis, si*PTPRK* and si*ZNRF3/RNF43* treatments both elevated LRP6 cell surface levels, but LRP6 levels were not further enhanced by their combined knockdown (*Figure 4E*). Likewise, depletion of PTPRK or ZNRF3/RNF43 elevated Topflash activity, while the combined knockdown did not further increase it (*Figure 4F* and *Figure 4—figure supplement 1E*).

Since a role for ZNRF3 has not been reported in *Xenopus*, we characterized its expression in *Xenopus tropicalis*. Maternal *znrf3* mRNA was detected in the animal hemisphere; in gastrulae it was prominently expressed in the organizer (*Figure 5A*), consistent with RNAseq analysis (*Figure 2D*). *ZNRF3* is a Wnt target gene (*Hao et al., 2012*) and likewise in *Xenopus* embryos it shows a pattern that follows high Wnt activity (*Figure 5A*) (*Borday et al., 2018*), including the posterior of early neurulae, and in tailbud embryos the midbrain, the dorsal neural tube and branchial arches.

We knocked down *znrf3* in *Xenopus* with two independent antisense Mos. One targets the splice site between exon 1 and intron 1 of *Xenopus tropicalis znrf3,* robustly reducing *znrf3* mRNA levels (Mo1, *Figure 5—figure supplement 1A*); the other targets the 5'-UTR (Mo2). Depletion of Znrf3 elicited axial defects that phenocopied *ptprk* morphants/crispants (*Figure 5B–C*). *Xenopus znrf3* morphants were rescued by coinjection of human untargeted *ZNRF3* mRNA (*Figure 5B–C*). Expectedly, *znrf3* Mo robustly induced Topflash activity in *Xenopus* embryos (*Figure 5—figure supplement 1B*). Both *ptprk* and *znrf3* show Spemann organizer expression and downregulate Wnt signaling. Accordingly, to examine whether Ptprk regulates Spemann organizer genes through Znrf3, we coinjected *ptprk* Mo with or without human *ZNRF3* mRNA. *ZNRF3* overexpression rescued both *gsc* and *chordin* expression, which were decreased by *ptprk* Mo (*Figure 5D* and *Figure 5—figure supplement 1C*). To test for their functional cooperation, we co-injected *ptprk* and *znrf3* antisense Mos at subthreshold doses, which individually hardly produced an effect. However, when combined, *ptprk* and *znrf3* Mos synergistically enhanced Topflash activity (*Figure 5E*). In addition, overexpression of human *ZNRF3* rescued Topflash induction by *ptprk* Mo (*Figure 5F*).

Taken together, the results support that PTPRK is an upstream positive regulator of ZNRF3 and thereby reduces cell surface Wnt receptors, which is essential for proper Spemann organizer function and *Xenopus* axial patterning.

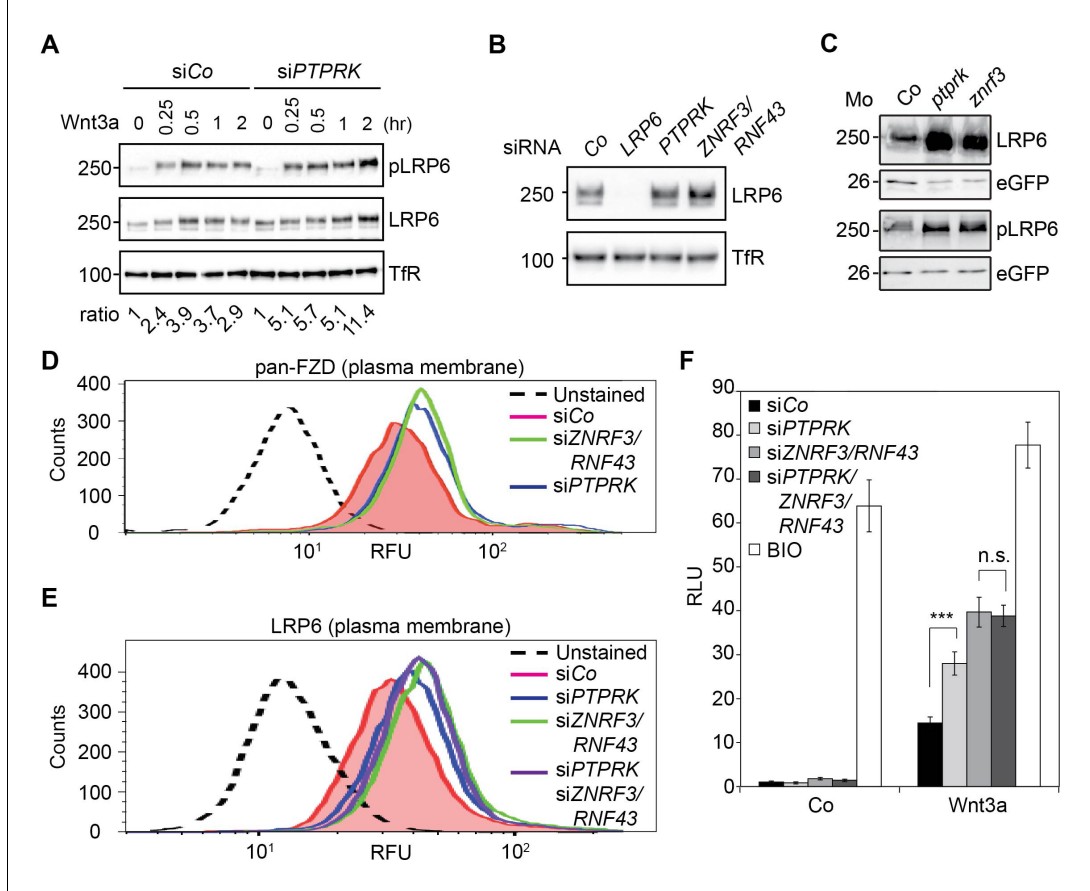

**Figure 4.** PTPRK reduces FZD and LRP6 surface levels via ZNRF3/RNF43. (**A**) Western blots analysis of membrane fractions from H1703 cells upon siRNA transfection. Cells were treated with Wnt3a for the indicated time and were analyzed. Ratio, phospho-LRP6 (pLRP6) levels normalized to control (transferrin receptor, TfR). Representative results from three independent experiments with similar outcome are shown. (**B**) Western blots analysis of membrane fractions from H1703 cells upon siRNA transfection. Transferrin receptor (TfR) served as loading control. Representative results from three independent experiments with similar outcome are shown. (**C**) Western blot analysis of LRP6 and phospho-LRP6 (pLRP6) in neurula (stage 18) embryos injected with *LRP6* RNA, *eGFP* RNA and indicated Mo. eGFP served as an injection control. Data show one representative result from three independent experiments. (**D**) Flow cytometric analysis of cell surface Frizzled receptors (pan-FZD antibody) in H1703 cells upon siRNA knockdown of PTPRK or ZNRF3/RNF43. Only live cells were counted and dead cells were gated out by propidium iodide (PI) staining. Dashed line, unstained H1703 cells. RFU, relative fluorescence units. (**E**) Flow cytometric analysis of cell surface LRP6 in H1703 cells upon siRNA depletion of PTPRK, ZNRF3/RNF43 or combination of both. Only live cells were counted and dead cells were gated out by propidium iodide (PI) staining. Dashed line, unstained H1703 cells. RFU, relative fluorescence units. (**F**) Topflash reporter assay in H1703 cells upon transfection of indicated siRNAs. Cells were treated with Wnt3a with or without 20 µM BIO for 24 hr before measurement. (Mean ± SD, n = 3; ***p<0.001, n.s., not significant, student t-test). RLU, relative light.
The online version of this article includes the following figure supplement(s) for figure 4:

**Figure supplement 1.** PTPRK regulates LRP6 protein but not mRNA levels.

## PTPRK promotes ZNRF3 mediated LRP6 and FZD degradation

We explored by co-immunoprecipitation (CoIP) if PTPRK and ZNRF3 physically interact. We used ZNRF3-ΔRING as it is more stable at the plasma membrane compared to wild-type ZNRF3. In CoIP experiments, full-length and phosphatase dead (DA) PTPRK bound to ZNRF3-ΔRING, whereas PTPRK-ΔC did not (*Figure 6A*). Moreover, PTPRK but not PTPRK-ΔC colocalized with ZNRF3-ΔRING in punctae at the plasma membrane (*Figure 6—figure supplement 1A*). These results indicate that PTPRK binds to ZNRF3 via its intracellular domain.

We generated a H1703 cell line harboring doxycycline (Dox) inducible ZNRF3-HA (TetOn ZNRF3-HA) to overcome both, poor transfection efficiency in this cell line and general lack of ZNRF3 antibodies. Employing this cell line, we tested if ZNRF3 is tyrosine phosphorylated and may be a substrate of PTPRK. By CoIP and Western blot detection with a phospho-Tyr-specific antibody, we

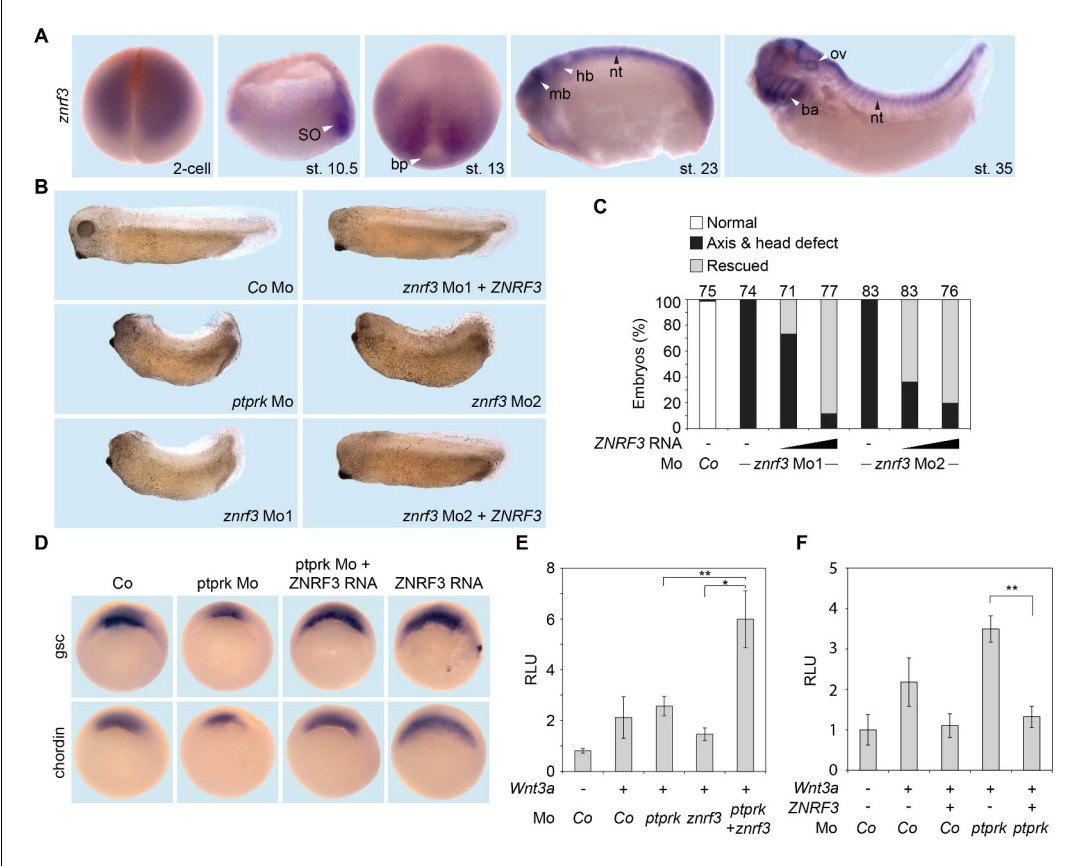

**Figure 5.** *Znrf3* is coexpressed- and cooperates with *ptprk* in early *Xenopus* embryos. (**A**) Spatial expression of *znrf3* in *Xenopus tropicalis* embryos at blastula (animal view), gastrula (hemisected dorsal to the right), neurula, tailbud and tadpole stages. ba, branchial arches; bp, blastopore; hb, hindbrain; mb, midbrain; nt, neural tube; ov, ovic vesicle; SO Spemann organizer. (**B**) Representative phenotypes of tailbud stage *Xenopus tropicalis* embryos injected animally at 2- to 8 cell stage as indicated. (**C**) Quantification of phenotypes shown in (**B**). The number of embryos per condition is indicated on the top. (**D**) Whole mount in situ hybridization of *gsc* and *chordin* in gastrula embryos (stage 10.5). Embryos were injected at 2- to 8 cell stage animally with *Co* or *ptprk* Mo with or without *ZNRF3* RNA. For quantification, see *Figure 5—figure supplement 1C*. (**E**) Topflash reporter assay performed with neurulae (stage 18). Embryos were injected animally at 2- to 8 cell stage as indicated. Suboptimal dosages of *ptprk* or *znrf3* Mos were used in this experiment. Normalized Topflash activity of *Co* Mo injected embryos was set to 1. (**F**) Topflash reporter assay performed with neurulae (stage 18). Embryos were injected animally at 2- to 8 cell stage as indicated. Normalized Topflash activity of *Co* Mo injected embryos was set to 1. Data in all graphs are displayed as means ± SD, and show one representative of multiple independent experiments with three biological replicates. RLU, relative light units. **p<0.01.

The online version of this article includes the following figure supplement(s) for figure 5:

**Figure supplement 1.** *Znrf3* Mo activates Wnt signaling in *Xenopus* embryos.

observed very little phosphorylated ZNRF3 (*Figure 6B*, lane 2). However, inhibiting endocytic traffic and lysosomal degradation with bafilomycin induced ZNRF3 phosphorylation, and treatment with the pan-PTP inhibitor Na-pervanadate (PV) massively increased ZNRF3 phosphorylation (*Figure 6B*, lanes 4, 6). These results suggest that i) ZNRF3 is tyrosine-phosphorylated but becomes rapidly dephosphorylated by PTPs, ii) that its phosphorylation status is related to vesicular traffic and lysosomal degradation. Interestingly, si*PTPRK* enhanced tyrosine phosphorylation of ZNRF3 both in control as well as in bafilomycin-treated cells (*Figure 6B*, lane 3, 5), suggesting that ZNRF3 is a substrate of PTPRK. Concordantly, when phosphorylated ZNRF3 was bound to immobilized PTPRK, ZNRF3 could be eluted by vanadate (*Figure 6C*), which mimics the conformation of the phosphate group at the transition state for phosphoryl transfer (*Lindquist et al., 1973*), hence indicating an enzyme-substrate interaction. Moreover, siRNA knockdown of other PTPRs also increased ZNRF3 phosphorylation, notably si*PTPRF* (*Figure 6—figure supplement 1B*), which also induced Wnt signaling (*AXIN2* expression; *Figure 1—figure supplement 1F*).

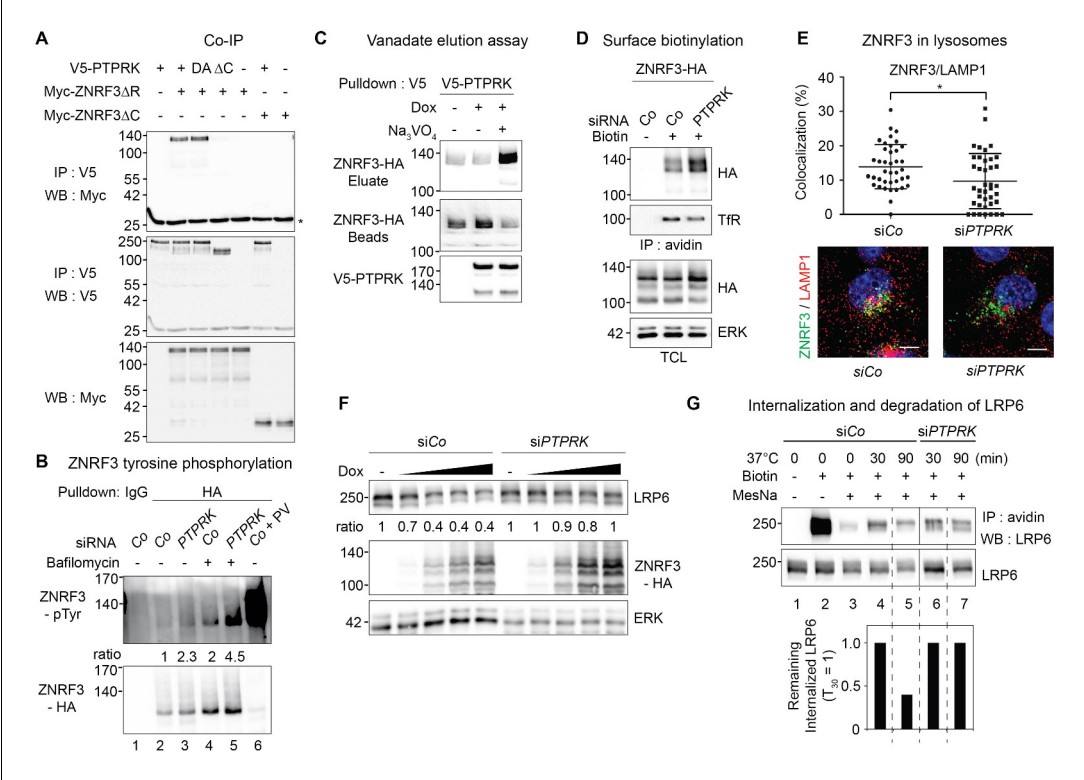

**Figure 6.** PTPRK binds ZNRF3 and promotes its dephosphorylation and lysosomal trafficking. (**A**) Co-immunoprecipitation experiments in HEK293T cells transfected with the indicated constructs and analyzed 48 hr after transfection. Data show a representative result from three independent experiments with similar outcomes. Asterisk, IgG light chain. (**B**) Tyrosine phosphorylation of ZNRF3 in TetOn ZNRF3-HA H1703 cells upon siRNA transfection with or without bafilomycin treatment overnight. Cells were treated with Dox for 48 hr before harvest. As a control, cells were treated with Na-pervanadate (PV, phosphatase inhibitor) for 30 min before harvest. Lysates were pulled down with anti-HA antibody or control IgG and subjected to Western blot analysis. Ratio, tyrosine phosphorylation of ZNRF3 normalized to total ZNRF3. (**C**) PTPRK-ZNRF3 interaction is vanadate-sensitive. Immobilized, immunoisolated V5-PTPRK from TetOn V5-PTPRK cells was incubated with total cell lysate from ZNRF3-HA expressing, Na-pervanadate treated cells. Bound ZNRF3-HA was eluted with 20 mM vanadate as indicated and eluate and beads were separated before Western blot analysis. (**D**) Cell surface biotinylation assay performed in TetOn ZNRF3-HA H1703 cells upon siRNA treatment. Cells were treated with Dox for 48 hr before harvest. After labeling surface proteins with Sulfo-NHS-LC-LC-Biotin, lysates were pulled down with streptavidin beads and subjected to Western blot analysis. Transferrin receptor (TfR), loading control for avidin pull down; ERK, total cell lysate (TCL) control. A representative result from three independent experiments with similar outcomes is shown. (**E**) Colocalization by immunofluorescence microscopy (IF) of ZNRF3 (Green) with LAMP1 (Red) in TetOn ZNRF3-HA H1703 cells upon siRNA treatment. Top, graph shows quantification of ZNRF3 colocalizing with LAMP1 (Mean ± SD, *p<0.05, student t-test). Bottom, representative IF images. Note that plasma membrane localized ZNRF3 cannot be seen due to low microscope laser power used for optimal vesicular co-localization in. Colocalization data are pooled from two independent experiments. (**F**) Western blot analysis of Dox treated TetOn ZNRF3-HA H1703 cells upon siRNA treatment. Ratio, LRP6 normalized to ERK. A representative result from three independent experiments with similar outcomes is shown. (**G**) Internalization and degradation assay of LRP6 in TetOn ZNRF3-HA H1703 cells treated as indicated. After labeling of surface proteins with Sulfo-NHS-SS-Biotin, endocytosis was induced by shifting cells to 37°C for the indicated times. At each indicated time point, cells were treated with MesNa to remove biotinylated surface proteins and then harvested. Cells were lysed, and biotinylated proteins were pulled down with streptavidin beads and analyzed with indicated antibodies. Lane 1: non-biotin treated control, Lane 3: MesNa treated after biotin labeling without inducing endocytosis (monitoring MesNa efficiency). The graph below shows avidin pulled down LRP6 levels normalized to total LRP6 levels (remaining internalized LRP6 at 30 min upon si*Co* was set to 1).

The online version of this article includes the following figure supplement(s) for figure 6:

**Figure supplement 1.** PTPRK regulates ZNRF3 trafficking but not its intrinsic E3 ligase activity.

ZNRF3 and RNF43 continuously degrade Wnt receptors by binding and recruiting them to the lysosome in an ubiquitin-dependent manner (*Koo et al., 2012*; *Tsukiyama et al., 2015*; *Park et al., 2018*). Hence, we analyzed whether PTPRK regulates ZNRF3 plasma membrane levels using a cell surface biotinylation assay. si*PTPRK* robustly increased surface levels of ZNRF3 but not that of ZNRF3-ΔRING (*Figure 6D* and *Figure 6—figure supplement 1C–D*), indicating that PTPRK promotes ZNRF3 internalization for which the RING domain is required. To analyze if tyrosine

phosphorylation impacts the E3 ligase activity of ZNRF3, we carried out an in vitro ubiquitination assay, monitoring autoubiquitination of ZNRF3 by using immunoprecipitated ZNRF3 and recombinant E2 ubiquitin conjugating enzyme. There was no change in ZNRF3 autoubiquitination following increased tyrosine phosphorylation upon either si*PTPRK* or Na-pervanadate treatment (*Figure 6— figure supplement 1E*). This suggests that tyrosine phosphorylation does not regulate the catalytic activity of ZNRF3.

We hypothesized that increased surface ZNRF3 upon PTPRK depletion is due to reduced lysosomal traffic. Concordantly, si*PTPRK* reduced the colocalization of ZNRF3 with the lysosomal marker LAMP1 (*Figure 6E*). In contrast, si*PTPRK* did not increase vesicular colocalization of ZNRF3 and Rab11 (recycling endosome marker) (*Figure 6—figure supplement 1F*).

ZNRF3 and RNF43 deplete Wnt receptors from the cell surface and target them towards lysosomal degradation (*Koo et al., 2012*; *Tsukiyama et al., 2015*). Consistently, in TetOn ZNRF3-HA cells, Dox treatment dose-dependently increased ZNRF3 and decreased LRP6 levels (*Figure 6F*). si*PTPRK* treatment reversed the effect on LRP6 and further increased ZNRF3 levels. Similarly, transfected ZNRF3 reduced FZD5 dose-dependently, while this effect of ZNRF3 was abolished upon si*PTPRK* treatment (*Figure 6—figure supplement 1G*). To confirm this result, we monitored the kinetics of LRP6 internalization and degradation using cleavable biotin. In si*Co* cells, internalized LRP6 was detected after 30 min (*Figure 6G*, compare lanes 2 and 4) and decreased after 90 min, likely due to lysosomal degradation (compare lanes 4 and 5). In contrast, si*PTPRK* prevented degradation of internalized LRP6 (compare lanes 6 and 7). Taken together, these results support a model in which vesicular trafficking of ZNRF3 and its ability to degrade Wnt receptors is regulated by tyrosine phosphorylation: Phosphorylation maintains plasma membrane residence while dephosphorylation by PTPRK promotes lysosomal targeting and degradation (*Figure 7—figure supplement 2*).

## PTPRK dephosphorylates a 4Y endocytic signal on ZNRF3

Tyrosine-containing motifs are known to play a critical role in regulating endocytosis of transmembrane proteins. Specifically, unphosphorylated YXXXφ, φXXY, as well as YXXφ (φ = bulky hydrophobic amino acid) sites can serve as internalization motifs (*Zhang and Allison, 1997*; *Roush et al., 1998*; *Bonifacino and Traub, 2003*; *Royle et al., 2005*). By multisequence alignment and inspection of the intracellular domain of ZNRF3, we identified a matching cluster of four adjacent tyrosine residues, or '4Y' motif (Y465, Y469, Y472 and Y473), which is highly conserved among vertebrates (*Figure 7A*). Each of these four tyrosine residues conforms to the aforementioned internalization motifs, suggesting that 4Y represents a cluster of four consecutive internalization signals.

To test whether the 4Y motif regulates ZNRF3 endocytosis, we designed a deletion construct ZNRF3(Δ4Y) (deletion of 9 amino acids encompassing the four tyrosines) and monitored its subcellular localization. Indeed, ZNRF3(Δ4Y) displayed enhanced membrane staining compared to wild-type (Wt) ZNRF3 (*Figure 7B*). Moreover, PTPRK knockdown induced tyrosine phosphorylation of Wt ZNRF3 but not that of ZNRF3(Δ4Y) (*Figure 7C*; compare lanes 3 and 5). This result was confirmed with a mutant ZNRF3(4YF), wherein all four tyrosine residues are substituted by phenylalanine (*Figure 7—figure supplement 1A*; compare lanes 3 and 7). Note though, that Na-pervanadate (PV) treatment induced massive tyrosine phosphorylation of ZNRF3 regardless of its mutation status, indicating additional PTPRK-independent phosphosites. We hypothesized that reduced endocytosis of ZNRF3(Δ4Y) would impair its ability to internalize Wnt receptors and render it hypoactive. Concordantly, ZNRF3(Δ4Y) downregulated FZD5 less efficiently than Wt ZNRF3 (*Figure 7D*). Moreover, ZNRF3(Δ4Y) and ZNRF3(4YF) were less efficient in inhibiting Topflash reporter assays compared to Wt ZNRF3 (*Figure 7E*; *Figure 7—figure supplement 1B*).

Taken together, our results suggest a model (*Figure 7—figure supplement 2*) where the 4Y motif of ZNRF3 represents an endocytic signal that promotes ZNRF3-Wnt receptor co-internalization. Phosphorylation of the 4Y motif by an unknown tyrosine kinase(s) prevents internalization and degradation of Wnt receptors, resulting in higher Wnt signaling. PTPRK counteracts this activity by dephosphorylating the 4Y motif, allowing efficient endocytosis of ZNRF3-Wnt receptor complexes and reducing Wnt signaling.

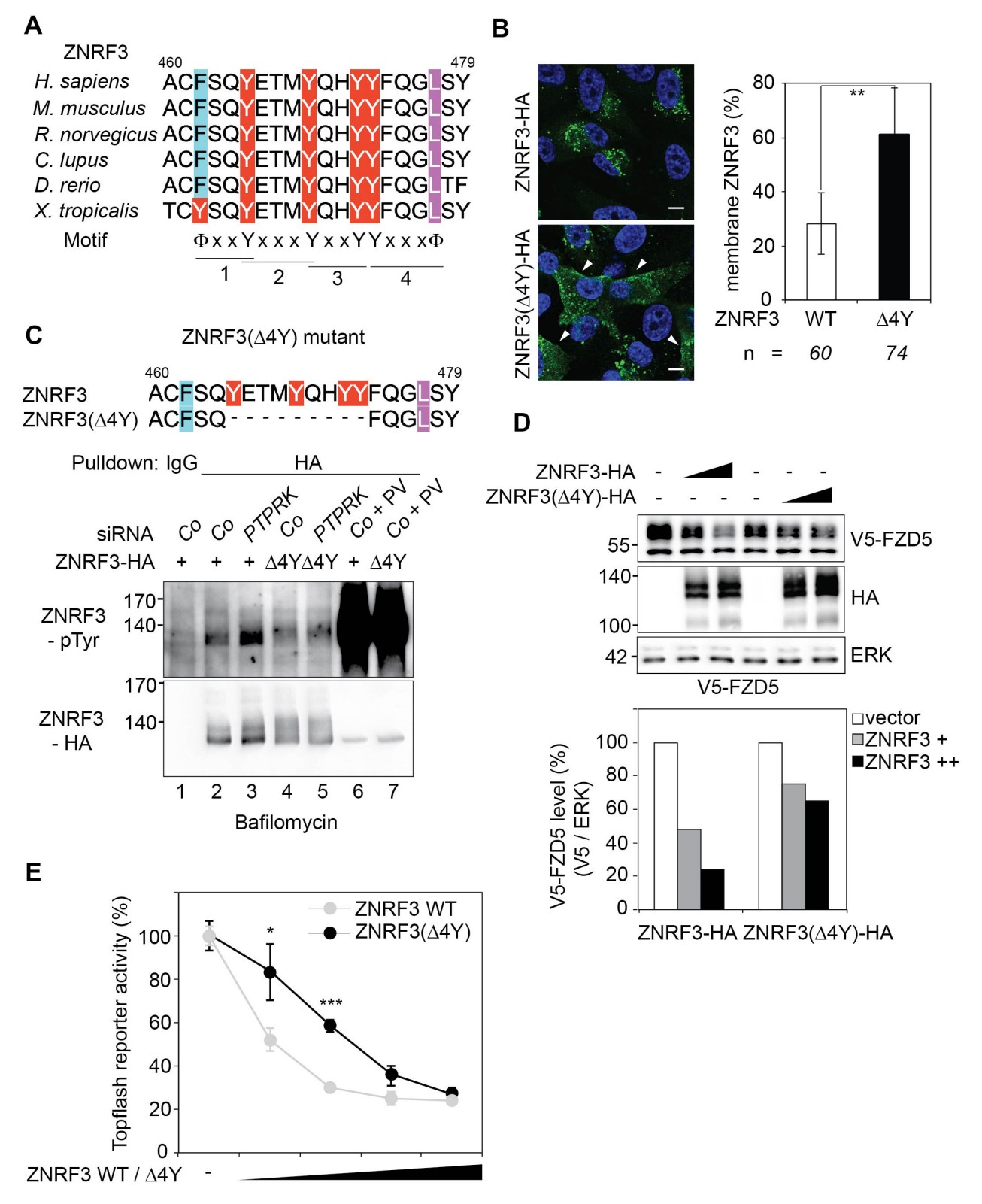

**Figure 7.** A "4Y" endocytic motif in ZNRF3 is regulated by PTPRK. (**A**) Multiple sequence alignment of ZNRF3 among different species. Y: Tyrosine, X: any amino acids, and Φ: hydrophobic bulky amino acids. (**B**) Subcellular localization by immunofluorescence microscopy (IF) of ZNRF3-HA or ZNRF3 (Δ4Y)-HA in H1703 cells with bafilomycin treatment overnight. Left, representative IF images. Arrowheads indicate membrane ZNRF3. Right, graph shows quantification of membrane ZNRF3 positive cells (Mean ± SD, **p<0.01, student t-test). The number of cells per condition is indicated at the

*Figure 7 continued on next page*

*Figure 7 continued*

bottom (n). IF data are pooled from two independent experiments. (C) Tyrosine phosphorylation of ZNRF3-HA or ZNRF3(Δ4Y)-HA in H1703 cells upon siRNA transfection with bafilomycin treatment overnight. As a control, cells were treated with Na-pervanadate (PV, phosphatase inhibitor) for 30 min before harvest. Lysates were pulled down with anti-HA antibody or control IgG and subjected to Western blot analysis. (D) Western blot analysis of H1703 cells transfected as indicated. The graph below shows quantification of V5-FZD5 normalized to ERK. The level of V5-FZD5 without ZNRF3-HA transfection was set to 100%. A representative result from two independent experiments with similar outcomes is shown. (E) Topflash reporter assay in H1703 cells upon transfection of different amount of ZNRF3-HA or ZNRF3(Δ4Y)-HA plasmids. All samples were Wnt3a treated for 24 hr before measurement. (Mean ± SD, n = 3; *p<0.05, ***p<0.001, student t-test). Topflash activity without ZNRF3 transfection was set to 100%. A representative result from three independent experiments with similar outcomes is shown.

The online version of this article includes the following figure supplement(s) for figure 7:

**Figure supplement 1.** The ZNRF3(4YF) mutant resists PTPRK regulation and shows reduced Wnt signal inhibition.
**Figure supplement 2.** Model for PTPRK acting as Wnt inhibitor.

## Discussion

The three main conclusions of this study are i) that the transmembrane phosphatase PTPRK, whose gene is found in prominent cancer-related fusion events with the ZNRF3 negative regulator *RSPO3*, is itself a positive regulator of ZNRF3. Thereby, PTPRK acts as negative regulator of Wnt/β-catenin signaling, enhancing Wnt receptor turnover; ii) that PTPRK depletes cell surface LRP6 and FZD by promoting lysosomal trafficking of ZNRF3, which it binds and whose tyrosine dephosphorylation on a 4Y endocytic signal it promotes; iii) that Wnt inhibition by PTPRK and ZNRF3 is essential in the Spemann organizer to regulate anterior neural development.

During animal development, Wnt signaling serves as a posteriorizing signal, and the tail-to-head gradient of Wnt activity is critical for the a-p specification of the neural plate (*Petersen and Reddien, 2009*; *Niehrs, 2010*). The Spemann organizer is a rich source of negative Wnt regulators, which maintain organizer function and promote anterior development. Joining this group of proteins, Ptprk is essential to downregulate Lrp6 and Wnt signaling to promote Spemann organizer and anterior development in *Xenopus*. Also the zebrafish *ptprk* ortholog is expressed in the early dorsal axis and notochord (*van Eekelen et al., 2010*). In contrast, *Ptrpk* null mutant mice are viable (*Skarnes et al., 1995*) and similarly we observed in mammalian cell-lines that the function of PTPRK is not universal but cell-line dependent, possibly reflecting redundancy with other RPTPs. Species differences in the essentiality of orthologous genes is common, even between the more closely related mouse and man, where > 20% of human essential genes have nonessential mouse orthologs (*Liao and Zhang, 2008*). Indeed, we found that PTPRF may also regulate ZNRF3 and Wnt signaling and be functionally redundant with PTPRK.

Despite its key importance as a negative Wnt regulator, the regulation of ZNRF3 is incompletely understood (*Deng et al., 2015*; *Shi et al., 2016*; *Ci et al., 2018*; *Qiao et al., 2019*). Our results in H1703 cells and *Xenopus* embryos clearly indicate that PTPRK regulates Wnt signaling in a phosphatase activity-dependent manner. Concordantly, PTPRK binds to ZNRF3 via its intracellular domain and the binding is abolished by vanadate, corroborating that ZNRF3 is a PTPRK substrate. We identify a 4Y endocytic signal in ZNRF3, which is tyrosine phosphorylated by an unknown kinase and dephosphorylated by PTPRK and whose mutation leads to plasma membrane accumulation of ZNRF3. Tyrosine phosphorylation is known to play an important role in sorting of transmembrane proteins to endosomes and lysosomes. For example, tyrosine phosphorylation of an endocytic YXXφ signal was shown to inhibit endocytosis and lysosomal targeting of CTLA-4 by decreasing binding to the endocytic adaptor protein AP2 (*Bonifacino and Traub, 2003*). Our results support a model in which lysosomal trafficking of ZNRF3 regulates its ability to degrade Wnt receptors, likely by escorting them (*Figure 7—figure supplement 2*). However, while PTPRK promotes ZNRF3 internalization via the 4Y motif, the overexpressed ZNRF3(Δ4Y) mutant is still able to deplete Wnt receptors and inhibit Wnt signaling, albeit less efficiently (*Figure 7D–E*). Hence, the 4Y motif and PTPRK only have a modulatory role towards ZNRF3.

PTPRK belongs to the R2B RPTP subfamily, which also includes PTPRM, PTPRT and PTPRU, sharing a common protein architecture (*Craig and Brady-Kalnay, 2015*). Among these, PTPRK, PTPRM, and PTPRT are implicated as tumor suppressors (*Zhao et al., 2010*; *Sudhir et al., 2015*), raising the possibility that they may also regulate ZNRF3 in certain cell types. We found that knockdown of

PTPRF (R2A) and PTPRH (R3) also increases ZNRF3 tyrosine phosphorylation in H1703 cells. PTPRK (R2B), PTPRF and PTPRS (R2A) were reported to dephosphorylate EGFR and attenuate EGF signaling (*Suárez Pestana et al., 1999*; *Xu et al., 2005*; *Wang et al., 2015*), suggesting that there can be functional redundancy between R2A and R2B RPTPs.

*PTPRK* is a candidate tumor suppressor in mouse intestinal tumorigenesis as per insertional mutagenesis (*Starr et al., 2009*; *March et al., 2011*), and is a gene fusion partner with the oncogene *RSPO3* in colorectal cancers (*Seshagiri et al., 2012*). Our results provide a rationale how PTPRK may function as a tumor suppressor in Wnt-ON tumors. ZNRF3 and RNF43 play a widespread role as negative feedback regulators in Wnt signaling (*Hao et al., 2012*). They are frequently mutated in a variety of cancers and their mutation signatures have shown promise as predictive biomarkers in pre-clinical models for the efficacy of upstream Wnt inhibitors (*Hao et al., 2016*). Downregulation of PTPRK and hence ZNRF3 would derepress Wnt-receptors, activate Wnt signaling, and promote tumorigenesis. However, other modes of action of PTPRK, such as dephosphorylating other signaling factors like β-catenin, EGFR, or STAT3 (*Xu et al., 2005*; *Novellino et al., 2008*; *Chen et al., 2015*), or cell junction proteins (*Fearnley et al., 2019*), may also contribute to its tumor-suppressive function.

Translocations where the signal sequence or part of the extracellular domain of PTPRK is fused to RSPO3 are recurrent events in a subset of colorectal cancers (*Seshagiri et al., 2012*). The tumorigenicity of these fusions has been solely attributed to upregulation of RSPO3 and hence ZNRF3/RNF43 depletion. However, our results indicate that haploinsufficiency of PTRPK could also contribute to tumorigenicity by further reducing ZNRF3 and increasing Wnt receptor levels. The close proximity of *PTPRK* and *RSPO3* loci and occurrence in gene fusions, their common function as Wnt signaling regulators, and the fact that at least one more Wnt regulatory gene, *RNF146* (*Zhang et al., 2011*), is located within the 1.5 Mb genomic interval encompassing *RSPO3* and *PTPRK*, is suggestive of a mini 'Wnt-operon' at this locus. Hence, it may be worthwhile probing the other four genes located in this interval (*ECHDC1, SOGA3, THEMIS, C6orf58*) for a Wnt-regulatory function.

Our results suggest that tyrosine kinases phosphorylating ZNRF3 at its 4Y endocytic signal are candidate targets for Wnt-directed tumor therapy, as their inhibition may promote ZNRF3 internalization and Wnt receptor turnover. Hence, it will be interesting in the future to characterize the 4Y kinase(s).

# Materials and methods

**Key resources table**

| Reagent type (species) or resource | Designation | Source or reference | Identifiers | Additional information |
|---|---|---|---|---|
| Gene (*Homo spiens*) | PTPRK | RZPD | DKFZ p686c2268Q2 | |
| Gene (*Homo spiens*) | ZNRF3 | Feng Cong PMID: 22575959 | | |
| Gene (*Homo spiens*) | FZD5 | NCBI | NM_003468.4 | |
| Strain, strain background (*Xenopus tropicalis*) | *Xenopus tropicalis* | Nasco | LM00822 | |
| Strain, strain background (*Xenopus tropicalis*) | *Xenopus tropicalis* | National Xenopus Resource (NXR) | NXR_1018 RRID:SCR_013731 | |
| Strain, strain background (*Xenopus tropicalis*) | *Xenopus tropicalis* | European Xenopus Resource Centre (EXRC) | RRID: SCR_007164 | |

*Continued on next page*

*Continued*

| Reagent type (species) or resource | Designation | Source or reference | Identifiers | Additional information |
|---|---|---|---|---|
| Cell line (*Homo-sapiens*) | H1703 | ATCC | CRL-5889 RRID: CVCL_1490 | |
| Cell line (*Homo-sapiens*) | 293T | ATCC | CRL-3216 RRID: CVCL_0063 | |
| Cell line (*Homo-sapiens*) | H1703 TetOn ZNRF3-HA | This paper | | generated from H1703 |
| cell line (*Homo-sapiens*) | H1703 TetOn V5-PTPRK | This paper | | generated from H1703 |
| Antibody | anti-transferrin receptor (Rabbit monoclonal) | Cell signaling | Cat# 13113 RRID:AB_2715594 | WB (1:5000) |
| Antibody | anti-LRP6 (Rabbit monoclonal) | Cell signaling | Cat# 2560 RRID:AB_2139329 | WB (1:1000) |
| Antibody | anti-LRP6 (Mouse monoclonal) | R and D systems | Cat#: MAP1505 RRID:AB_10889810 | FACS (2.5 µg/ml) |
| Antibody | anti-phospho LRP6 (Sp1490) (Rabbit monoclonal) | Cell signaling | Cat#: 2568 RRID:AB_2139327 | WB (1:1000) |
| Antibody | anti-Rab11 (Mouse monoclonal) | BD bioscience | Cat#: 610656 RRID:AB_397983 | IF (1:200) |
| Antibody | anti-β-catenin (Mouse monoclonal) | BD bioscience | Cat#: 610154 RRID:AB_397555 | WB (1:5000) |
| Antibody | anti-phosphotyrosine (4G10) (Mouse monoclonal) | BD bioscience | Cat#: 610000 RRID:AB_397423 | IF (1:1000) |
| Antibody | anti-Erk1/2 (Rabbit polyclonal) | Sigma Aldrich | Cat#: M8159 RRID:AB_477245 | WB (1:5000) |
| Antibody | anti-V5 (Mouse monoclonal) | Thermo scientific | Cat#: R960-25 RRID:AB_2556564 | WB (1:5000) IF (1:1000) |
| Antibody | anti-HA (Rat polyclonal) | Roche | Cat#: 1867423 RRID:AB_390918 | WB (1:1000) IF (1:1000) |
| Antibody | anti-Myc (Mouse monoclonal) | DSHB | Cat#: 9E10 RRID:AB_2266850 | WB (1:1000) |
| Antibody | anti-Myc (Rabbit polyclonal) | Millipore | Cat#: 06–549 RRID:AB_310165 | IF (1:1000) |
| Antibody | anti-pan FZD (humanized) | Austin Gurney (former Oncomed pharmaceuticals) | OMP-18R5 | FACS (2.5 µg/ml) |
| Antibody | anti-GFP (Rabbit polyclonal) | Invitrogen | Cat#: A11122 RRID:AB_221569 | WB (1:1000) |

*Continued on next page*

*Continued*

| Reagent type (species) or resource | Designation | Source or reference | Identifiers | Additional information |
|---|---|---|---|---|
| Antibody | anti-Ubiquitin (Rabbit polyclonal) | Dako | Cat#: Z0458 RRID:AB_2315524 | WB (1:1000) |
| Antibody | anti-LAMP1 (Rabbit polyclonal) | Abcam | Cat#: ab24170 RRID:AB_775978 | IF (1:200) |
| Antibody | goat-anti mouse HRP | Dianova | 115-035-174 RRID:AB_2338512 | WB (1:10000) |
| Antibody | goat-anti rabbit HRP | Dianova | 111-035-144 RRID:AB_2307391 | WB (1:10000) |
| Antibody | goat-anti rat HRP | Dianova | 112-035-175 RRID:AB_2338140 | WB (1:10000) |
| Antibody | goat-anti mouse Alexa 488 | Invitrogen | A11029 RRID:AB_138404 | IF (1:200) |
| Antibody | donkey-anti rabbit Alexa 647 | Invitrogen | A31573 RRID:AB_2536183 | IF (1:200) |
| Antibody | goat-anti rat Alexa 488 | Invitrogen | A11006 RRID:AB_141373 | IF (1:200) |
| Antibody | donkey-anti mouse Alexa 647 | Invitrogen | A31571 RRID:AB_162542 | IF (1:200) |
| Antibody | goat-anti human Alexa 488 | Invitrogen | A11013 RRID:AB_141360 | FACS (1:500) |
| Recombinant DNA reagent | pCS2+ (plasmid) | Ralph AW Rupp PMID: 7926732 | | |
| Recombinant DNA reagent | pCS2-V5PTPRK (plasmid) | This paper | | See 'Expression constructs' |
| Recombinant DNA reagent | pCS2-V5PTPRK D1057A (plasmid) | This paper | | See 'Expression constructs' |
| Recombinant DNA reagent | pCS2-V5PTPRK C1089S (plasmid) | This paper | | See 'Expression constructs' |
| Recombinant DNA reagent | pCS2-V5PTPRK ΔC (plasmid) | This paper | | See 'Expression constructs' |
| Recombinant DNA reagent | pcDNA4/TO-ZNRF3HA (plasmid) | Feng Cong PMID: 22575959 | | |
| Recombinant DNA reagent | Myc-ZNRF3ΔRING (plasmid) | Feng Cong PMID: 22575959 | | |
| Recombinant DNA reagent | pcDNA4/TO-ZNRF3(Δ4Y) (plasmid) | This paper | | See 'Expression constructs' |
| Recombinant DNA reagent | pcDNA4/TO-ZNRF3(4YF) (plasmid) | This paper | | See 'Expression constructs' |
| Recombinant DNA reagent | pcDNA4/TO-ZNRF3-Flag (plasmid) | Feng Cong PMID: 25891077 | | |
| Recombinant DNA reagent | pCS2-V5-Frizzled5 | This paper | | See 'Expression constructs' |
| Recombinant DNA reagent | pCDNA3-mWnt1-Myc | from Xi He | | |

*Continued on next page*

*Continued*

| Reagent type (species) or resource | Designation | Source or reference | Identifiers | Additional information |
|---|---|---|---|---|
| Recombinant DNA reagent | pRK5-mFz8 | from J Nathans | | |
| Recombinant DNA reagent | pCS2-hLRP6 | from Xi He | | |
| Recombinant DNA reagent | pCS2-hDvl1 | RZPD | IRALp962D1142 | |
| Recombinant DNA reagent | hβ-catenin S37A | from M Boutros | | |
| Peptide, recombinant protein | Catalase | Sigma Aldrich | Cat. #: C1345 | |
| Commercial assay or kit | NucleoSpin RNA | Macherey-Nagel | Cat. #: 740955 | |
| Commercial assay or kit | SuperSignal West pico ECL | Thermo Scientific | Cat. #: 34577 | |
| Chemical compound, drug | Bafilomycin | Calbiochem | Cat. #: 196000 | |
| Chemical compound, drug | BIO | Cayman chemical company | Cat. #: 16329 | |
| Chemical compound, drug | Mesna | Cayman chemical company | Cat. #: 21238 | |
| Software, algorithm | LightCycler 480 software | Roche | 4994884001 | |
| Software, algorithm | Fluoroskan Ascent FL software | Thermo scientific | 11540775 | |
| Software, algorithm | LAS 3000 Reader ver 2.2 | Fuji film | | |
| Software, algorithm | Multi-gauge ver 3.2 | Fuji film | | |
| Software, algorithm | FlowJo software ver 10.5.3 | BD | RRID:SCR_008520 | |
| Software, algorithm | Zen black | Carl Zeiss | RRID:SCR_013672 | |
| Software, algorithm | Graphpad | Prism | RRID:SCR_002798 | |
| Software, algorithm | Fiji (image J) | Open source PMID: 22743772 | RRID:SCR_002285 | |
| Other | Hoechst | Sigma Aldrich | B-2883 | (1 µg/mL) |
| Other | Protein A magnetic bead | Thermo Scientific | 88846 | |
| Other | Strepavidin agarose | Thermo Scientific | 20359 | |
| Other | sulfo-NHS-LC-LC-Biotin | Thermo Scientific | 21338 | 0.25 mg/ml |
| Other | sulfo-NHS-SS-Biotin | Thermo Scientific | 21331 | 0.5 mg/ml |

## Cell culture

H1703 cells (ATCC) were maintained in RPMI with 10% FBS, supplemented with 2 mM L-glutamine, 1 mM sodium pyruvate and penicillin/streptomycin. HEK293T cells (ATCC) were maintained in DMEM with 10% FBS, supplemented with 2 ml L-glutamine and 1 mM penicillin/streptomycin. Cell identity was authenticated by ATCC by STR profiling. Regular mycoplasma test showed both cell lines were mycoplasma negative.

## Expression constructs

V5 tagged PTPRK (1–1446), PTPRKΔC (1-776) and FZD5 (1–585) were generated by inserting human PTPRK or FZD5 into a pCS-based vector containing the V5 epitope after the signal peptide of mouse *Krm2*. Site directed mutagenesis for V5-PTPRK-CS (C1089S; Catalytic Cys in phosphate binding site changed to Ser) and V5-PTPRK-DA (D1057A; Asp in WPD loop changed to Ala) was done by two-step PCR and mutations were validated by sequencing. hZNRF3-HA, Myc-ZNRF3-ΔRING and Myc-ZNRF3-ΔC were kindly provided by F. Cong (*Hao et al., 2012*). ZNRF3(Δ4Y)-HA (deletion of 465–474) or ZNRF3(4YF)-HA (phenylalanine substitution of Y465, Y469, Y472 and Y473) were done by amplification of whole plasmids with 5'-phosphorylated primers followed by DpnI digestion and self-ligation.

## Real time quantitative PCR

H1703 cells in 12-well plates were lysed with RNA lysis buffer containing 1% β-mercaptoethanol. RNA isolation was performed with Nucleospin RNA isolation kit following the manufacturer's instruction (Macherey-Nagel, Düren, Germany). Reverse transcription and PCR amplification were performed as described before (*Berger et al., 2017*). Primers and siRNA information are listed in *Supplementary file 1*.

## Luciferase reporter assay

For Topflash assay in H1703 cell line, $3.25 \times 10^3$ cells per well were plated in 96-well plates. Where indicated, cells were transfected with 25 nM siRNAs using Dharmafect (Dharmacon, Lafayette, CO). After 24 hr, cells were transfected with plasmids including 5 ng of pTK-Renilla, 25 ng of SuperTop, 2 ng of mWnt1, 2.4 ng of hLRP6, 0.24 ng of Mesd, 0.8 ng of mFzd8, 12 ng of hDvl1, 0.08 ng of human β-catenin S37A using Lipofectamine 3000 (Invitrogen, Carlsbad, CA). pCS2+ vector was used to adjust total DNA amount to 100 ng per well. For Topflash assay in HEK293T cell line, $10^4$ cells per well were plated in 96-well plates and 1 ng of pTK-Renilla, 5 ng of SuperTop, were transfected. After 48 hr of DNA transfection, luciferase activities were measured with Dual-luciferase kit (Promega, Madison, WI). When necessary, Wnt3a conditioned medium was treated 24 hr before measuring the luciferase activities.

## Western blot and immunoprecipitation

For isolation of total cell lysates, cells were harvested in cold PBS and lysed with Triton lysis buffer (20 mM Tris-Cl, pH 7.5, 1% Triton X-100, 150 mM NaCl, 1 mM EDTA, 1 mM EGTA, 1 mM β-glycero-phosphate, 2.5 mM sodium pyrophosphate, 1 mM Na-orthovanadate) supplemented with complete protease inhibitor cocktail (Roche, Basel, Switzerland). For membrane-enriched fractions, cells were lysed with Saponin lysis buffer (20 mM Tris-Cl, pH 7.5, 0.05% Saponin, 1 mM $MgCl_2$, 1 mM Na-ortho-vanadate) supplemented with complete protease inhibitor cocktail (Roche, Basel, Switzerland). After centrifugation, the supernatant (cytosolic fraction) was discarded and the pellets were lysed with Tri-ton lysis buffer. Lysates were cleared by centrifugation, and Bradford assay was performed to measure the protein concentration. For Western blot, 30 µg of lysates were mixed with NuPage LDS sample buffer containing 50 mM DTT and heated at 70°C for 10 min.

For co-immunoprecipitation or pull-down assay, 300 ~ 800 µg of total cell lysates were precleared with 10 µl of A/G plus agarose (Santacruz Biotechnologies, Santacruz, CA) on a rotator at 4°C for 1 hr. Precleared lysates were incubated with 10 µl of anti-V5 agarose (Sigma Aldrich, St. Louis, MO) or with 20 µl of A/G plus agarose with anti-HA (1867423; Roche, Basel, Switzerland) on a rotator at 4°C overnight. Immunoprecipitated proteins were washed with triton lysis buffer for four times and mixed with NuPage LDS sample buffer containing 50 mM DTT, followed by heated at 70°C for 10 min. Samples were subjected to SDS-PAGE, transferred to nitrocellulose membrane, and blocked

with 5% BSA in TBST (10 mM Tris-Cl, pH 8.0, 150 mM NaCl, 0.05% Tween-20). Primary antibodies in blocking buffer were applied overnight at 4°C, and incubation of secondary antibodies was carried out at RT for 1 hr. Western blot images were taken with SuperSignal West pico ECL (Thermo Scientific, Waltham, MA) using LAS-3000 (Fujifilm, Tokyo, Japan). Densitometry analyses were done with Multi-gauge software (Fujifilm, Tokyo, Japan). Antibody information is listed in key resource table.

## Vanadate elution assay

TetOn V5-PTPRK or V5-PTPRK-DA H1703 cells (bait) were harvested after 48 hr of doxycycline (200 ng/ml) treatment and lysed in 400 µl lysis buffer A (20 mM Tris-Cl, pH 7.5, 100 mM NaCl, 10% Glycerol, 1% Triton) supplemented with complete protease inhibitor cocktail (Roche, Basel, Switzerland). One milligram of total cell lysate was pulled down with 300 ng anti-V5 antibody plus 10 µl protein A magnetic beads (88846; Thermo Scientific, Waltham, MA) overnight. Beads were washed twice with lysis buffer A and mixed with prey (see below).

TetOn ZNRF3-HA H1703 cells (prey) were harvested after treatment of 100 µM of freshly prepared Na-pervanadate for 30 min and washed twice with cold PBS followed by lysis with 400 µl buffer B (50 mM Tris-Cl, pH 7.5, 150 mM NaCl, 10% Glycerol, 1% Triton, 1 mM EDTA, pH 8.0, 5 mM Iodoacetamide and 10 mM NaF). One milligram of total cell lysate was added to the immunoprecipitated bait on protein A beads with additional 500 µl lysis buffer A for 2 hr at 4°C with rotation. After four times wash with lysis buffer A, vanadate elution was done at RT for 30 min by adding 20 mM Na-orthovanadate in 30 µl of buffer A. Supernatants were separated from beads and both were boiled with LDS sample buffer with 50 mM DTT and subjected to SDS-PAGE.

For the preparation of Na-pervanadate, 10 µl of 100 mM $Na_3VO_4$ was added to 50 µl of 20 mM HEPES (pH 7.5) containing 0.3% $H_2O_2$, followed by 940 µl $H_2O$ and 5 min incubation. 2 µg of catalase (C1345; Sigma Aldrich, St. Louis, MO) was added for 5 min to remove unreacted $H_2O_2$.

## Subcellular fractionation

H1703 cells were harvested in cold PBS 72 hr post siRNA transfection, and lysed with hypotonic buffer (5 mM HEPES, 1 mM $MgCl_2$, 2 mM Na-orthovanadate) containing complete protease inhibitor cocktail (Roche, Basel, Switzerland). After 30 min of incubation on ice, cell membranes was disrupted by syringe pipetting with 26 gauge needles, followed by centrifugation at 800 x g for 5 min at 4°C. The supernatant was centrifuged at 5,000 rpm for 5 min at 4°C to remove the debris. After further centrifugation at 30,000 rpm for 20 min at 4°C using a Beckman TL-100 with TLA-55 rotor, the pellet contained the membrane fraction and the supernatant the cytosolic part. The membrane pellet was solubilized in Triton lysis buffer. Equal amounts of proteins were mixed with NuPAGE LDS Sample Buffer, heated at 70°C for 10 min, followed by SDS-PAGE and Western blot analysis.

## In vitro ubiquitination assay

TetOn ZNRF3-HA H1703 cells were seeded in 10 cm dishes ($4.3 \times 10^5$ cells per dish) and transfected with the indicated siRNAs. After 24 hours, cells were treated with doxycycline (200 ng/ml) to activate *ZNRF3* expression. Three days post induction cells were harvested and lysed in 400 µl Triton lysis buffer (50 mM Tris-Cl, pH 7.5, 150 mM NaCl, 10% Glycerol, 1% Triton, 1 mM EDTA, pH 8.0, 5 mM Iodoacetamide, 1 mM Na-orthovanadate, 10 mM N-Ethylmaleimide and 10 mM NaF). After preclearing the lysates with A/G plus agarose for 1 hr at 4°C, they were pulled down with 150 ng anti-HA and 20 µl A/G plus agarose for 4 hr at 4°C followed by four washes with lysis buffer (20 mM Tris-Cl, pH 7.5, 100 mM NaCl, 10% Glycerol, 1% Triton) and once with PBS. The ZNRF3 IP-beads were resuspended in a volume of 10 µl containing reaction buffer (40 mM HEPES, pH 7.4, 50 mM NaCl, 8 mM magnesium acetate), 10 µM Ubiquitin, 30 µM ATP, 50 nM UBE1 (E1), 2 µM UbcH5b (E2) as indicated in the Figure. Samples were incubated for 5 hr at 37°C with gentle shaking before boiling in NuPAGE LDS Sample Buffer containing 50 mM DTT for 2 min at 95°C, followed by PAGE analysis.

## Immunofluorescence microscopy

Cells were grown on coverslips in 6-well plates and fixed in 4% PFA for 10 min. The immunofluorescence experiments were performed as published (*Berger et al., 2017*). Coverslips were mounted with Fluoromount G.

## Flow cytometry analysis

Cells were harvested with Versane solution (Lonza, Basel, Switzerland) and washed with FACS buffer (PBS, 1% BSA, 0.1% Sodium Azide) followed by blocking with FACS buffer containing 20 μl of FcγR inhibitor (eBioscience, San Diego, CA) for 30 min. After blocking, samples were incubated with 2.5 μg/ml of pan-FZD or LRP6 antibody at 4°C for 3 hr followed by two washes with FACS buffer. Goat anti-human Alexa488 or goat anti-mouse Alexa488 with a dilution of 1:1000 was applied to the sample for 1 hr at 4°C. After two washes with FACS buffer, samples were incubated with 1 μg/ml of propidium iodide for 5 min before analysis on a FACScalibur. Ten thousand live cells per sample were acquired and analyzed with FlowJo (Tree Star Inc, Ashland, OR).

## Cell surface biotinylation assay

H1703 cells were transfected with siRNA for 72 hr and the washed three times with cold PBS. Surface proteins were biotinylated with 0.25 mg/ml sulfo-NHS-LC-LC-Biotin (Thermo scientific, Waltham, MA) at 4°C for 30 min. For non-biotinylated control, PBS was added. The reaction was quenched by 3 washes with 10 mM Monoethanolamine and cells were harvested and lysed with Triton lysis buffer. 200–300 μg of lysate was incubated with 10 μl streptavidin agarose (Thermo scientific, Waltham, MA) to pull-down biotinylated surface proteins, and precipitated proteins were subjected to Western blot and detected with indicated antibodies.

## Surface internalization assay with cleavable sulfo-NHS-SS-Biotin

TetOn ZNRF3-HA H1703 cells were transfected with siRNA for 24 hr and then treated with doxycycline (200 μg/ml) for 48 hr. Surface proteins were biotinylated with 0.5 mg/ml sulfo-NHS-SS-Biotin (Thermo Scientific, Waltham, MA) at 4°C for 30 min. After quenching excessive biotin with 10 mM Monoethanolamine, pre-warmed culture medium was added for the indicated times at 37°C to induce internalization. At the indicated times, remaining surface-biotin was removed by 50 mM MesNa (2-mercaptoethanesulfonate, membrane impermeable reducing agent) in 100 mM Tris-HCl, pH 8.6, 100 mM NaCl and 2.5 mM $CaCl_2$ at 4°C for 30 min and MesNa protected biotinylated proteins were analyzed. Cells were lysed with RIPA buffer (20 mM Tris-Cl, pH 7.4, 120 mM NaCl, 1% Triton X-100, 0.25% Na-deoxycholate, 0.05% SDS, 50 mM sodium fluoride, 5 mM EDTA, 2 mM Na-orthovanadate) supplemented with complete protease inhibitor cocktail (Roche, Basel, Switzerland). 200–300 μg lysate was incubated with 10 μl streptavidin agarose (Thermo Scientific, Waltham, MA) to pull down biotinylated protein, and precipitated proteins were subjected to Western blot and detected with indicated antibodies.

## *Xenopus* methods

*Xenopus tropicalis* frogs were obtained from Nasco, National Xenopus Resource (NXR) and European Xenopus Resource Centre (EXRC). In vitro fertilization, embryo culture, preparation of mRNA, and microinjection were carried out as described (*Gawantka et al., 1995*). For *Xenopus tropicalis* embryo injection, mRNA/DNA/Morpholino oligonucleotide (Mo) was injected animally between the 2- to 8 cell stage. Equal amounts of total mRNA/DNA or Mo were injected by adjustment with pre-prolactin (*PPL*) RNA/DNA or standard control Mo (GeneTools, Philomath, OR). Based on *Xenopus tropicalis ptprk* sequence (ENSXETG00000010633), an antisense Mo was designed: 5'-TTCTTACC TGCACACTTGGTTCTTG-3'. The sequence of the antisense Mo targeting *Xenopus tropicalis znrf3* (ENSXETG00000019942) was: 5'-CCACTTACCTGCACGATCTCCCCCT-3' (Mo1, splice-blocking Mo) and 5'-AACATAATTTCCCAGTCCTCAGTGG-3' (Mo2, translation-blocking Mo). Injected amount (per embryo) of each Mo was as follows: 0.5 or 1 ng of *lrp6* Mo, 1, 2, or 5 ng of *β-catenin* Mo, and 5 or 10 ng of *ptprk* Mo, 2 or 10 ng of *znrf3* Mo1, 40 ng of *znrf3* Mo2. The injected mRNA amounts were 1 pg *Wnt3a*, 500 pg *PTPRK* WT or mutants, and 30 pg *ZNRF3*.

For luciferase reporter assays, embryos were injected with Topflash and Renilla-TK plasmid DNA plus indicated Mos and synthetic mRNA. Three pools of 5 embryos each were lysed with passive lysis buffer (Promega, Madison, WI) and assayed for luciferase activity using the Dual luciferase system (Promega, Madison, WI). All luciferase reporter assays represent the mean ± standard error of 3 independent measurements of pools (five embryos per pool; total n = 15 per experiment shown). The reproducibility was confirmed by at least three independent experiments in different batches of *Xenopus tropicalis* embryos.

Whole-mount in situ hybridizations were carried out essentially as described (*Gawantka et al., 1995*). The in situ hybridization probe for *Xenopus tropicalis ptprk* was generated by PCR using *Xenopus tropicalis ptprk* cDNA (IMAGE ID: 7708108) as a template, a forward primer: 5′-CCCCCCGGGGAGCCTCCAAGGCCTATTGC-3′, and a reverse primer: 5′-CCCGAATTCGGATGGTAGTCCCTGGATGC-3′ to amplify a fragment with a size of 835 bp. The PCR product was cloned into pBluescript KS+ using SmaI and EcoRI as the upstream and downstream cloning site respectively. The in situ hybridization probe for *Xenopus tropicalis znrf3* was generated by PCR using cDNA (IMAGE ID: 7656097) as template, a forward primer: 5′-ATAAGAATGCGGCCGCATGCACCCACTTGGACTCTGTAAT −3′, and a reverse primer: 5′-ACGCGTCGACGTCCTGAAGATGCATGGTCCAGT-3′ to amplify a fragment with a size of 1000 bp. The PCR product was cloned into pBluescript KS+ using NotI and SalI. For lineage tracing, embryos were injected with 10 ng of *ptprk* Mo or 10 pg of *Wnt8* DNA plus *lacZ* mRNA (200 pg per embryo). Embryos were collected at embryonic stage 11 (gastrula) or 18 (neurula) and processed for in situ hybridization. β-galactosidase staining was performed as described (*Bradley et al., 1996*) using Rose-Gal substrate (Genaxxon bioscience, Ulm, Germany). Phenotypes were scored using a stereomicroscope by comparing wild-type and Mo-injected embryo morphology and counting embryos with the indicated abnormalities.

For animal cap assay, embryos were injected at 2- to 8 cell stage with 100 pg (per embryo) of *noggin* RNA and indicated Mos into the animal hemisphere. Animal cap explants were excised at stage 9 from 20 embryos and cultivated in 0.5x Barth solution containing Penicillin/Streptomycin. Animal cap explants were harvested at stage 18 and lysed in TRIzol (Thermo scientific, Waltham, MA) for RNA extraction, and qRT-PCR assays were performed to analyze the expression of indicated genes.

For qRT-PCR analysis, 10 embryos at tailbud stage (stage 25) or 20 animal cap explants at neurula stage equivalent (stage 18) were harvested and lysed in 1 ml of TRIzol (Thermo Scientific, Waltham, MA), and RNA extraction and precipitation was performed following the manufacturer's instruction. Reverse transcription was performed with 1 μg RNA using SuperScript II reverse transcriptase and random primers (Invitrogen). The obtained cDNA was subjected to PCR amplification using UPL (Universal ProbeLibrary; Roche, Basel, Switzerland) probes and corresponding primers, and analyzed by LightCycler 480 (Roche, Basel, Switzerland).

For Western blot analysis, *Xenopus tropicalis* embryos were injected with indicated Mos at 2- to 8 cell stage into the animal hemisphere. Embryos were harvested at stage 18, homogenized in NP-40 lysis buffer (2% NP-40, 20 mM Tris-HCl, pH 7.5, 150 mM NaCl, 10 mM NaF, 10 mM $Na_3VO_4$, 10 mM sodium pyrophosphate, 5 mM EDTA, 1 mM EGTA, 1 mM PMSF, and protease inhibitors (Roche, Basel, Switzerland) with a volume of 4 μl per embryo. Lysates were cleared with Freon followed by centrifugation (21,000 x g, 10 min at 4°C), 70°C for 10 min with NuPAGE LDS Sample Buffer, and SDS-PAGE analysis.

CRISPR/Cas9-mediated mutagenesis was performed as described (*Nakayama et al., 2014*). In brief, embryos were injected at one-cell in the animal hemisphere with 5 nl per embryo. After injection, embryos were cultured in 1/18 MR until stage 18 for Luciferase assays or stage 30 for phenotyping and genotyping. The putative sgRNA target site for *Xenopus tropicalis ptprk* and specificity check were predicted on online database CCTop - CRISPR/Cas9 target online predictor (https://crispr.cos.uni-heidelberg.de/) (*Stemmer et al., 2015*) and CRISPRdirect (https://crispr.dbcls.jp/) (*Naito et al., 2015*) using the exon 1 sequence of *ptprk* (Transcript ID: ENSXETT00000023302.3). The linear DNA template for *ptprk* sgRNA was synthesized using a PCR-based strategy. The 5′ primer was: 5′-GCAGCTAATACGACTCACTATAGTGTGGTGGTGCAATAGGCCTGTTTTAGAGCTAGAAATA-3′, and the 3′ primer was: 5′-AAAAGCACCGACTCGGTGCCACTTTTTCAAGTTGATAACGGACTAGCCTTATTTTAACTTGCTATTTCTAGCTCTAAAAC-3′. For genotyping using restriction enzyme digestion, individual embryo was transferred to a 0.2 ml PCR tubes containing 100 μl of lysis buffer (50 mM Tris, pH 8.8, 1 mM EDTA, 0.5% Tween 20,) with freshly added proteinase K at a final concentration of 200 μg/ml. Embryos were incubated at 56°C for 2 hr to overnight, followed by 95°C for 10 min to inactivate proteinase K. Lysates were centrifuged at 17,000 x g for 10 min at 4°C. One microliter of lysate was used as a template for PCR to amplify the targeted genomic region using a forward primer: 5′-AGCCTCAGTCTGGCTTTTTAATTT-3′, and a reverse primer: 5′-CTCAAGGTTAACGCTACGAAAAATC-3′. The PCR products were digested by StuI and analyzed by agarose electrophoresis.

## Acknowledgements

We thank C Cruciat, D Ingelfinger, and M Boutros for cooperation during the original siRNA screen, and Y Cheng for providing *Cas9* plasmid; F Cong (Novartis) for providing the ZNRF3 constructs; A Gurney for providing pan-FZD antibody (OMP-18R5; vantictumab); H Ulrich and C Renz for providing materials and advice for the in vitro ubiquitination assay. We thank Dr. G Roth and Aska Pharmaceuticals Tokyo for generous supplying hCG. Expert technical support by the DKFZ core facility for flow cytometry, light microscopy and the central animal laboratory of DKFZ is gratefully acknowledged. We thank A Hirth for help with BioRender. This work was supported by the DFG (CRC 1324).

## Additional information

### Funding

| Funder | Grant reference number | Author |
| --- | --- | --- |
| Deutsche Forschungsgemeinschaft | CRC1324 | Ling-Shih Chang<br>Minseong Kim<br>Andrey Glinka<br>Carmen Reinhard<br>Christof Niehrs |

The funders had no role in study design, data collection and interpretation, or the decision to submit the work for publication.

### Author contributions

Ling-Shih Chang, Minseong Kim, Conceptualization, Validation, Investigation, Visualization, Writing - original draft, Writing - review and editing; Andrey Glinka, Resources, Methodology, Contributed to interpretation of data; Carmen Reinhard, Resources, Validation; Christof Niehrs, Conceptualization, Supervision, Funding acquisition, Writing - original draft, Project administration, Writing - review and editing

### Author ORCIDs

Ling-Shih Chang ⓘ https://orcid.org/0000-0003-0025-0862
Minseong Kim ⓘ https://orcid.org/0000-0002-3927-4899
Christof Niehrs ⓘ https://orcid.org/0000-0002-9561-9302

### Ethics

Animal experimentation: All Xenopus experiments were approved by the state review board of Baden-Wuerttemberg, Germany (License number: G-13/186 & G-141/18) and performed according to the federal and institutional guideline.

### Decision letter and Author response

Decision letter https://doi.org/10.7554/eLife.51248.sa1
Author response https://doi.org/10.7554/eLife.51248.sa2

## Additional files

### Supplementary files

• Supplementary file 1. Morpholinos, siRNA and primers used in this study.

• Transparent reporting form

### Data availability

All data generated or analysed during this study are included in the manuscript.

The following previously published dataset was used:

**Database and**

| Author(s) | Year | Dataset title | Dataset URL | Identifier |
|---|---|---|---|---|
| Ding Y, Colozza G, Zhang K, Moriyama Y, Ploper D, Sosa EA, Benitez MDJ, De Robertis EM | 2016 | Genome-wide analysis of dorsal and ventral transcriptomes of the Xenopus laevis gastrula | https://www.ncbi.nlm.nih.gov/geo/query/acc.cgi?acc=GSE75278 | NCBI Gene Expression Omnibus, GSE75278 |

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
