## [Decision Letter]

**Acceptance summary:**

The work in this paper centers around the function and expression of Wnt antagonists in the Spemann organizer. Several of these antagonists had been described in the past but the authors identify a novel activity, the protein tyrosine phosphatase receptor-type kappa (PTPRK). This protein was previously known to be acting as a tumor suppressor. Moreover, the authors link the activity of PTPRK to another Wnt inhibitor, the transmembrane E3 ubiquitin ligase ZNRF3, which was implicated in Wnt receptor degradation. ZNRF3 is also expressed in the Spemann organizer. Inactivation of PTPRK increases Wnt signaling and induces head and axial developmental defects. Mechanistically, PTPRK interacts with a '4Y' endocytic signal on ZNRF3 molecule. The finding of PTPRK acting as an inhibitor of Wnt receptor turnover may explain its action as a tumor suppressor.

**Decision letter after peer review:**

Thank you for submitting your article "The tumor suppressor PTPRK promotes ZNRF3 function and is required for Wnt inhibition in the Spemann organizer" for consideration by *eLife*. Your article has been reviewed by three peer reviewers, one of whom is a member of our Board of Reviewing Editors and the evaluation has been overseen by Michael Eisen as the Senior Editor. The reviewers have opted to remain anonymous.

The reviewers have discussed the reviews with one another and the Reviewing Editor has drafted this decision to help you prepare a revised submission.

Your paper "The tumor suppressor PTPRK promotes ZNRF3 function and is required for Wnt inhibition in the Spemann organizer" has now been reviewed by three referees. As you will see, the reviewers' opinions are mixed. Each of them expresses interest in the data and comment in positive words on the relevance of your findings for understanding the role of Wnt signaling in development and in cancer.

On the other hand, two of the reviewers ask for a fairly obvious addition: an analysis and discussion of the localization of tyrosine residues within the ZNRF3 tail. It seems to me that you must have considered this and possibly have done the experiments. If added to the paper, it would strengthen the evidence that ZNRF3 phosphorylation is increased by *siPTPRK*. It would then also possibly show that these residues are conserved and whether mutations of those residues generate a hyperactive ZNRF3 protein.

The reviewers raise a number of other issues that you should consider when submitting a revised version. Reviewer 2 suggests to adjust the interpretation of the results.

*Reviewer #1:*

The authors have identified the tumor suppressor Protein tyrosine phosphatase receptor-type kappa (PTPRK), as a Wnt inhibitor. Most of the work is done in *Xenopus*, as part of investigations into the nature of the organizer.

Previously, it was known that PTPRK can be part of a fusion protein, with the Wnt modulator RSPO3 in cancer fusions. The new work would now suggest that both fusion partners, PTPRK and RSPO3 are contributing to excess Wnt activity in tumors.

A major new result in the manuscript is that PTPRK acts via the transmembrane E3 ubiquitin ligase ZNRF3, a negative feedback regulator of Wnt signaling. The authors claim that ZNRF3 is tyrosine phosphorylated and that PTPRK binds to ZNRF3 and promotes its dephosphorylation and internalization.

The results would explain that a deficiency of PTPRK would lead to an increase in Wnt signaling, consequently leading to reduced expression of Spemann organizer effector genes and defects in axis formation.

The work is overall interesting and well done. There is however one piece of information missing, as surely realized by the authors as well: what are the phosphorylated tyrosine residues on ZNRF3? This is important to know, as it may also shed light on the nature of the tyrosine kinase phosphorylating ZNRF3. The phosphorylated tyrosine sites could themselves be targets in cancer as well. It seems to me that it should be possible to experimentally establish which sites are phosphorylated.

A related question is whether the binding of PTPRK to ZNRF3, as shown in figure 6A, is dependent on the phosphorylated status of ZNRF3.

Reviewer #2:

This study reports on the identification of PTPRK as a negative regulator of Wnt signaling. Using Wnt-responsive H1703 cells and *Xenopus* embryos combined with siRNA and CRISPR-based approaches, the inhibitory effects of PTPRK are positioned upstream in the Wnt cascade and shown to depend on phosphatase activity. Depletion of PTPRK mediates increased LRP6 and FZD levels at the cell surface of both H1703 cells and neurula embryos. These findings suggest that PTPRK acts by negatively regulating Wnt receptor levels at the cell surface. Due to phenotypic similarities with the Wnt receptor-downregulating Ub ligases ZNRF3 and RNF43, the authors investigate a cooperative mode of action between PTPRK and ZNRF3. Mechanistically, the authors propose a model in which PTPRK enhances Wnt receptor turnover via dephosphorylation of ZNRF3, followed by its enhanced lysosomal trafficking and turnover.

Although the identification of PTPRK as a negative regulator of Wnt signaling is of potential interest, the claims made on the underlying mechanism appear not sufficiently substantiated. The evidence that ZNRF3 phosphorylation is increased by *siPTPRK* is rather weak and consequences for ZNRF3 internalization and lysosomal turnover appear inconsistent. Direct dephosphorylation of ZNRF3 by PTPRK is not shown. In addition, alternative explanations of the results are not explored.

Major points:

1) Alternative possibilities are not considered – can the authors exclude a direct effect of PTPRK on the regulation of Wnt receptor stabilization?

2) An analysis and discussion of the localization of tyrosine residues within the ZNRF3 tail is lacking. Are these residues conserved? Are they located within or close to known internalization motifs? Which of those Tyr residues show increased phosphorylation in PTPRK-depleted cells? Do mutation of those Tyr residues generate a hyperactive ZNRF3 protein? Insights in these issues will help to understand how Tyr phosphorylation might prevent ZNRF3 endocytosis and trafficking.

3) No data are shown on how PTPRK might regulate RNF43, the functional homologue of ZNRF3. Are the effects of PTPRK selective for ZNRF3?

4) The fact that overexpression of PTPRK does not show a phenotype is puzzling and not well-explained by the model. What about expression of a catalytically inactive variant, are any dominant negative effects seen?

5) Based on the results presented in Figure 6B, the authors state that an increase in ZNRF3 phosphorylation in PTPRK-depleted cells corresponds with enhanced lysosomal turnover (Figure 6B). However, levels of phosphorylation merely seem to increase due to stabilization of ZNRF3 protein in bafilomycin-treated cells. Also, these results should be shown with an empty lane in between lane 5 and 6 to prevent spilling over due to the strongly increased levels of phosphorylation in PV-treated cells (lane 6).

6) In the co-IP experiments of Figure 6C, ZNRF3 ends up in the pellet even though no V5-PTPRK is expressed and thus no material should be pulled down (lane 1). Thus, ZNRF3 is expected to reside in the supernatant fraction in these conditions.

7) In Figure 6D, surface levels of ZNRF3 appear increased upon PTPRK depletion. Do these cell surface-retained receptors contain higher pTyr levels? This will be important to show, to substantiate the point that Tyr phosphorylation prevents endocytic trafficking.

8) In Figure 6E, PTPRK kd decreases the colocalization of ZNRF3 with LAMP1. Surprisingly, no increase at the plasma membrane is observed, while this would be expected based on the results shown in 6D. Also, as ZNRF3 does not end up in LAMP1-positive late endosomes/lysosomes in PTPRK-depleted cells (Figure 6E), in what intracellular compartments does the protein reside?

9) ZNRF3 phosphorylation does not appear to affect ubiquitin ligase activity. In many models, ubiquitination actually follows phosphorylation and destines proteins for endocytosis. The endocytic system is loaded with Ub-binding adaptor proteins. How do the authors envision that ubiquitinated ZNRF3 species are prevented from being internalized? How does the interplay between ZNRF3 phosphorylation and ubiquitination take place?

Reviewer #3:

This is an interesting paper that adds to our understanding of Wnt regulation in diverse contexts, though developmental and tumorigenesis settings are of particular interest here.

There is already an abundance of negative regulation of Wnt signaling, with recent interest in the turnover of receptors by the Ubiqutin E3 ligases ZNRF3 and RNF43. In turn, these E3 ligases are turned over by R-spondin/lgr5 so that Wnt signaling is potentiated. In this current paper, an additional layer of regulation of Wnt receptors by a Transmembrane Phosphatase has been demonstrated. The PTPRK phosphatase removes phosphates from the ZNRF3 molecule, stimulating internalization, where it is proposed to carry the Fzd/LRP6 receptor complexes into the lysosome. Of interest to the tumor field, there are gene fusions between the adjacent PTPRK and RSPO3 that increase RSPO3 expression. Normally RSPO3 binds to Lgr5 and stimulates removal of ZNFR3/RNF43, thereby allowing the Wnt receptors to stay on the cell surface. The fusion also reduces the activity of PTPRK, so phosphorylated ZNRF3 no longer internalizes and destabilizes FZD/LRP6, also leading to increased Wnt signaling. Thus, a new finding is that the gene fusion is predicted to have dual effects, both of which potentiate Wnt signaling, and promote tumor formation.

Figure 1. The effect was discovered in an RNAi screen, and followed up carefully with Topflash, Axin2 expression, immunofluorescence, and Western blot tests for the function and abundance of β-catenin.

Figure 2. shows the expression of the PTPRK gene in frog embryos, where it is enriched in the organizer and other tissues. Knockdown by splice blocking Mo or CRISPR causes a small head and truncated tail phenotype, typical of excess Wnt signaling, and TOPflash experiments confirm this.

Figure 3 shows effects on organizer gene expression of the *ptprk* Mo, presumably because this interferes with the early cleavage Wnt signal that promotes organizer formation, in a secondary effect at the gastrula and neurula stage, the Mos cause restriction of anterior gene expression.

Figure 4 shows that knocking down the E3 ligases or PTPRK the amount of Fzd or LRP6 on the cell surface.

Figure 5 shows expression of znfr3 in coincident locations to ptprk in *Xenopus* embryos, and that both knockdowns work in a similar direction.

Figure 6 shows biochemical interaction between ZNRF3 and PTPRK, and the consequences of knockdown of PTPRK on the subcellular distribution of ZNRF3, with more efficient removal of ZNFR3 to lysozomes in the presence of PTPRK.

Supplementary figures show appropriate controls quantifying various effects.

The supplemental summary figure is useful, though it might be a little clearer to show another (dephosphorylated) ZNRF3 molecule on its way to interact with the FZD LRP6 complex. The work is clearly presented, though with so many negative interactions, a little mind bending.

---

## [Author Response]

Your paper "The tumor suppressor PTPRK promotes ZNRF3 function and is required for Wnt inhibition in the Spemann organizer" has now been reviewed by three referees. As you will see, the reviewers' opinions are mixed. Each of them expresses interest in the data and comment in positive words on the relevance of your findings for understanding the role of Wnt signaling in development and in cancer.On the other hand, two of the reviewers ask for a fairly obvious addition: an analysis and discussion of the localization of tyrosine residues within the ZNRF3 tail. It seems to me that you must have considered this and possibly have done the experiments. If added to the paper, it would strengthen the evidence that ZNRF3 phosphorylation is increased by siPTPRK. It would then also possibly show that these residues are conserved and whether mutations of those residues generate a hyperactive ZNRF3 protein.The reviewers raise a number of other issues that you should consider when submitting a revised version. Reviewer 2 suggests to adjust the interpretation of the results.

We appreciate the editors and the reviewer’s constructive suggestions, which helped improving our manuscript. We have now addressed all the comments, including by new experiments. Notably, as requested, we now have identified the tyrosine phosphorylation site in ZNRF3 that is regulated by PTPRK. It is an evolutionary conserved endocytic signal, consisting of four consecutive tyrosine residues and conforming to classical internalization signal motifs. Mutation of the site eliminates PTPRK regulation and affects ZNRF3/Wnt signaling. We believe this finding has significantly improved our study, opening up further possibilities to specifically interfere with ZNRF3 function e.g. in cancer context.

In addition to requested experiments, we added one unsolicited experiment to the manuscript, which is displayed in Figure 5D and Figure 5 —figure supplement 1C. The data show that ZNRF3 overexpression can restore the decrease of Spemann organizer gene expression caused by ptprk removal. This further strengthens our model that Ptprk inhibits Wnt signaling via Znrf3 in Spemann organizer.

Reviewer #1:The authors have identified the tumor suppressor Protein tyrosine phosphatase receptor-type kappa (PTPRK), as a Wnt inhibitor. Most of the work is done in Xenopus, as part of investigations into the nature of the organizer.Previously, it was known that PTPRK can be part of a fusion protein, with the Wnt modulator RSPO3 in cancer fusions. The new work would now suggest that both fusion partners, PTPRK and RSPO3 are contributing to excess Wnt activity in tumors.A major new result in the manuscript is that PTPRK acts via the transmembrane E3 ubiquitin ligase ZNRF3, a negative feedback regulator of Wnt signaling. The authors claim that ZNRF3 is tyrosine phosphorylated and that PTPRK binds to ZNRF3 and promotes its dephosphorylation and internalization.The results would explain that a deficiency of PTPRK would lead to an increase in Wnt signaling, consequently leading to reduced expression of Spemann organizer effector genes and defects in axis formation.The work is overall interesting and well done. There is however one piece of information missing, as surely realized by the authors as well: what are the phosphorylated tyrosine residues on ZNRF3? This is important to know, as it may also shed light on the nature of the tyrosine kinase phosphorylating ZNRF3. The phosphorylated tyrosine sites could themselves be targets in cancer as well. It seems to me that it should be possible to experimentally establish which sites are phosphorylated.

It is known that Tyrosine-containing YXXφ, YXXXφ, and φXXY sites can serve as internalization motifs (where φ denotes a bulky hydrophobic amino acid; Royle et al., 2005; Roush et al., 1998). In the cytoplasmic domain of ZNRF, we identified a matching cluster of four adjacent tyrosines, or ‘4Y’ motif (Y465, Y469, Y472 and Y473), which is highly conserved in vertebrates (Figure 7A). We constructed ZNRF3 mutants with deletion (ZNRF3(Δ4Y)) or Phe-substitution (ZNRF3(4YF)) of the 4Y motif. Unlike for Wt ZNRF3, PTPRK knockdown does not increase tyrosine phosphorylation of either ZNRF3(Δ4Y) or ZNRF3(4YF) (Figure 7A, C and Figure 7—figure supplement 1A). These results indicate that PTPRK regulates tyrosine phosphorylation via the 4Y motif on ZNRF3 to regulate endocytosis of ZNRF3.

A related question is whether the binding of PTPRK to ZNRF3, as shown in figure 6A, is dependent on the phosphorylated status of ZNRF3.

No, binding of PTPRK to ZNRF3 is independent on the phosphorylated status of ZNRF3 when tested with the new ZNRF3 mutants (Author response 1). This result seems inconsistent with Figure 6C, which shows that PTPRK bound to ZNRF3 is eluted by Na_3_VO_4_. The explanation for this discrepancy could lie in a recent finding of Fearnley et al., (2019). They reported that 30% of all PTPRK substrates bind to its non-catalytic D2 domain, indicating that this distal domain provides additional substrate specificity in addition to the catalytic D1 phosphatase domain. We speculate that when PTPRK and ZNRF3 are co-transfected for co-IP, their binding occurs mainly via the D2 domain due to limited amount of endogenous kinases phosphorylating the tyrosine cluster. In contrast, in the vanadate elution experiment, we treated ZNRF3 expressing cells with Na-pervanadate prior to harvest, which massively induces tyrosine phosphorylation. Under these conditions, binding between PTPRK and ZNRF3 would be more dominated by phosphorylation- dependent interaction via the D1 phosphatase domain.

**Author response image 1. respfig1:** Internalization-motif mutants of ZNRF3 still bind PTPRK. CoIP experiments in HEK293T cells transfected with the indicated constructs. Data show a representative result from three independent experiments with similar outcomes.

Reviewer #2:This study reports on the identification of PTPRK as a negative regulator of Wnt signaling. Using Wnt-responsive H1703 cells and Xenopus embryos combined with siRNA and CRISPR-based approaches, the inhibitory effects of PTPRK are positioned upstream in the Wnt cascade and shown to depend on phosphatase activity. Depletion of PTPRK mediates increased LRP6 and FZD levels at the cell surface of both H1703 cells and neurula embryos. These findings suggest that PTPRK acts by negatively regulating Wnt receptor levels at the cell surface. Due to phenotypic similarities with the Wnt receptor-downregulating Ub ligases ZNRF3 and RNF43, the authors investigate a cooperative mode of action between PTPRK and ZNRF3. Mechanistically, the authors propose a model in which PTPRK enhances Wnt receptor turnover via dephosphorylation of ZNRF3, followed by its enhanced lysosomal trafficking and turnover.Although the identification of PTPRK as a negative regulator of Wnt signaling is of potential interest, the claims made on the underlying mechanism appear not sufficiently substantiated. The evidence that ZNRF3 phosphorylation is increased by siPTPRK is rather weak and consequences for ZNRF3 internalization and lysosomal turnover appear inconsistent. Direct dephosphorylation of ZNRF3 by PTPRK is not shown. In addition, alternative explanations of the results are not explored.Major points:1) Alternative possibilities are not considered – can the authors exclude a direct effect of PTPRK on the regulation of Wnt receptor stabilization?

We have investigated the possibility of LRP6 as a substrate of PTPRK. First, we tested for increased tyrosine phosphorylation of LRP6 upon PTPRK knockdown. However, at least in H1703 cells, we could not detect tyrosine phosphorylation of LRP6 without Na-pervanadate treatment (Author response image 2). Second, as shown in Figure 4F and 4E, PTPRK knockdown does not change Topflash or surface LRP6 level when ZNRF3/RNF43 are depleted. Third, PTPRK knockdown stabilizes both LRP6 and Frizzled (Figure 4D-E), which imitates loss of ZNRF3/RNF43, supporting that PTPRK regulates Wnt receptors via ZNRF3/RNF43.

**Author response image 2. respfig2:** PTPRK depletion does not induce LRP6 Tyrosine phosphorylation. Tyrosine phosphorylation of LRP6 in H1703 cells with Wnt3a treatment upon siRNA transfection. As a control, cells were treated with Na-pervanadate (PV, phosphatase inhibitor) for 30 min before harvest. Cells were treated with Wnt3a for the indicated time before harvest. Lysates were pulled down with anti-LRP6 antibody or control IgG and subjected to Western blot analysis.

2) An analysis and discussion of the localization of tyrosine residues within the ZNRF3 tail is lacking. Are these residues conserved? Are they located within or close to known internalization motifs? Which of those Tyr residues show increased phosphorylation in PTPRK-depleted cells? Do mutation of those Tyr residues generate a hyperactive ZNRF3 protein? Insights in these issues will help to understand how Tyr phosphorylation might prevent ZNRF3 endocytosis and trafficking.

It is known that Tyrosine-containing YXXφ, YXXXφ, and φXXY sites can serve as internalization motifs (where φ denotes a bulky hydrophobic amino acid; Royle et al., 2005; Roush et al., 1998). In the cytoplasmic domain of ZNRF, we identified a matching cluster of four adjacent tyrosines, or ‘4Y’ motif (Y465, Y469, Y472 and Y473), which is highly conserved in vertebrates (Figure 7A). We constructed ZNRF3 mutants with deletion (ZNRF3(Δ4Y)) or Phe-substitution (ZNRF3(4YF)) of the 4Y motif. Unlike for Wt ZNRF3, PTPRK knockdown does not increase tyrosine phosphorylation of either ZNRF3(Δ4Y) or ZNRF3(4YF) (Figure 7A, C and Figure 7—figure supplement 1A). These results indicate that PTPRK regulates tyrosine phosphorylation via the 4Y motif on ZNRF3 to regulate endocytosis of ZNRF3.

A hyperactive ZNRF3 would require Tyr substitution by a constitutively unphosphorylated Tyr-mimic, which to our knowledge does not exist (Tyr in YXXφ cannot be substituted by other aromatic amino acids; Bonifacino and Traub, 2003). Instead, both deletion and phenylalanine mutation of the tyrosine based endocytic motif are predicted to disrupt the internalization motif of ZNRF3, and hence to render the protein hypoactive. Indeed, ZNRF3(Δ4Y) degraded FZD5 less efficiently than wild-type ZNRF3 (Figure 7D). Moreover, transfected ZNRF3(Δ4Y) showed prominent membrane localization compared to wild-type ZNRF3 (Figure 7B), supporting that the 4Y motif functions as internalization signal and is essential for efficient degradation of Wnt receptors. In line with this, ZNRF3(Δ4Y) and ZNRF3(4YF) showed reduced ability to inhibit Topflash activity compared to Wt ZNRF3 (Figure 7E and Figure 7—figure supplement 1B).

3) No data are shown on how PTPRK might regulate RNF43, the functional homologue of ZNRF3. Are the effects of PTPRK selective for ZNRF3?

Unlike ZNRF3, close inspection by multi-sequence alignment showed that RNF43 does not contain the 4Y motif, nor any other conserved Tyr residues in an internalization consensus context. Next, we tested whether PTPRK modulates tyrosine phosphorylation of RNF43. We monitored pTyr in transfected RNF43 upon PTPRK knockdown and found tyrosine phosphorylation of RNF43 was unaffected by PTPRK depletion (Author response image 3). Moreover, we found that Na-pervanadate treatment does not induce massive tyrosine phosphorylation of RNF43 like in ZNRF3 (Figure 7C, lane 6, 7).

**Author response image 3. respfig3:** si*PTPRK* does not affect tyrosine phosphorylation of RNF43. (**A**) Table showing the comparison of multiple sequence alignment result of ZNRF3 and RNF43 intracellular domain among different species including *H. sapiens, M. musculus, R. norvegicus, C. lupus, D. rerio* and *X. tropicalis*. (**B**) si*PTPRK* does not affect tyrosine phosphorylation of RNF43. Tyrosine phosphorylation of RNF43-FLAG in H1703 cells was analyzed in bafilomycin treated cells. As a control, cells were treated with Na-pervanadate (PV, phosphatase inhibitor) for 30 min before harvest. Lysates were pulled down with anti-FLAG antibody or control IgG and subjected to Western blot analysis. A representative result from two independent experiments with similar outcomes is shown.

4) The fact that overexpression of PTPRK does not show a phenotype is puzzling and not well-explained by the model. What about expression of a catalytically inactive variant, are any dominant negative effects seen?

As shown in our Figure 1—figure supplement 1D, we have transfected PTPRK and PTPRKDA (catalytically inactive PTPRK) cells to test the effect of overexpression of PTPRK on Wnt signaling. However, we have not observed any significant effect on Topflash activity. The lack of gof activity might be due to the need of PTPRK to undergo proteolytic processing by multiple proteases for full activity (Anders et al., 2006). These proteolytic steps may be limiting when PTPRK is overexpressed.

5) Based on the results presented in Figure 6B, the authors state that an increase in ZNRF3 phosphorylation in PTPRK-depleted cells corresponds with enhanced lysosomal turnover (Figure 6B). However, levels of phosphorylation merely seem to increase due to stabilization of ZNRF3 protein in bafilomycin-treated cells. Also, these results should be shown with an empty lane in between lane 5 and 6 to prevent spilling over due to the strongly increased levels of phosphorylation in PV-treated cells (lane 6).

We do not exclude that the increase of tyrosine-phosphorylated ZNRF3 by PTPRK knockdown might also be partially due to overall ZNRF3 stabilization. However, when pTyr is normalized to total ZNRF3, there is still more phosphorylation upon PTPRK knockdown (Figure 6B).

Regarding spill-over: in Figure 7C and Figure 7—figure supplement 1A, we loaded Na-pervanadate treated samples two lanes or five lanes away from PTPRK knockdown samples respectively, and we could still observe increased phosphorylation of ZNRF3 upon PTPRK removal.

6) In the co-IP experiments of Figure 6C, ZNRF3 ends up in the pellet even though no V5-PTPRK is expressed and thus no material should be pulled down (lane 1). Thus, ZNRF3 is expected to reside in the supernatant fraction in these conditions.

The experiment in Figure 6C is not a regular Co-IP. As bait, we pulled down V5-PTPRK from Dox-inducible cells. As prey, ZNRF3 was prepared from another inducible cell line under Na-pervanadate treatment, which massively induces ZNRF3 phosphorylation. Then, we mixed bait and prey and eluted bound ZNRF3 with 20 mM Na_3_VO_4_. This eluate is what we indicated as Sup and the beads are referred to as Pellet. Thus, ZNRF3 cannot be seen in supernatant from lane 1 because it is already washed out, and the ZNRF3 signal from pellet or sup is background. We realized that this misunderstanding is due to insufficient explanation for the experiment and have clarified this point.

7) In Figure 6D, surface levels of ZNRF3 appear increased upon PTPRK depletion. Do these cell surface-retained receptors contain higher pTyr levels? This will be important to show, to substantiate the point that Tyr phosphorylation prevents endocytic trafficking.

To answer this question, we fractionated samples into nuclear/ER-, membrane- and cytosol fractions, followed by pulldown of ZNRF3 and detection with phospho-tyrosine antibody (Author response image 4). Tyrosine phosphorylation of ZNRF3 was clearly increased upon PTPRK knockdown in the membrane fraction, while there was no change in the nuclear/ER fraction, supporting that PTPRK regulates phosphorylation of ZNRF3 at the plasma membrane level, consistent with our model.

**Author response image 4. respfig4:** PTPRK regulates tyrosine phosphorylation of ZNRF3 at the plasma membrane. Left: Tyrosine phosphorylation of ZNRF3 in TetOn ZNRF3-HA H1703 cells upon siRNA transfection after subcellular fractionation. As a control, cells were treated with Na-pervanadate (PV, phosphatase inhibitor) for 30 min before harvest. Right: Subcellular markers to validate fractionation. Lamin B: Nucleus; TfR (transferrin receptor), Plasma membrane; Tubulin, cytosol marker. Note that nuclear fraction displaying TfR signal indicates that nuclear fraction contains ER. Asterisk: non-specific bands from Na-pervanadate treatment. A representative result from two independent experiments with similar outcomes is shown.

8) In Figure 6E, PTPRK kd decreases the colocalization of ZNRF3 with LAMP1. Surprisingly, no increase at the plasma membrane is observed, while this would be expected based on the results shown in 6D. Also, as ZNRF3 does not end up in LAMP1-positive late endosomes/lysosomes in PTPRK-depleted cells (Figure 6E), in what intracellular compartments does the protein reside?

Imaging was optimized for vesicular ZNRF3 detection. ZNRF3 staining is much stronger in vesicles than at the plasma membrane. With higher imaging sensitivity, we now show increased ZNRF3 staining on the plasma membrane following PTPRK knockdown (Figure 6—figure supplement 1D).

The reviewer also asked in which intracellular compartment ZNRF3 resides beside lysosome. Colocalization with organelle markers indicates that ZNRF3 punctae co-localize with TGNP (trans-Golgi network) and Rab7a (late endosome) (Author response image 5).

**Author response image 5. respfig5:** PTPRK resides in secretory vesicles and endosomes. Immunofluorescence microscopy showing colocalization of ZNRF3-HA (green) with mCherry-Calreticulin, mCherry-TGNP or mCherry-Rab7a (red).

9) ZNRF3 phosphorylation does not appear to affect ubiquitin ligase activity. In many models, ubiquitination actually follows phosphorylation and destines proteins for endocytosis. The endocytic system is loaded with Ub-binding adaptor proteins. How do the authors envision that ubiquitinated ZNRF3 species are prevented from being internalized? How does the interplay between ZNRF3 phosphorylation and ubiquitination take place?

We are monitoring auto-ubiquitination of ZNRF3, which was unaffected by phosphorylation status, suggesting that phosphorylation does not induce a conformational change, which would affect the catalytic activity of the ZNRF3 ligase. This is different from what the referee refers to, where other substrate proteins are ubiquitinated by E3 ligases, which is often phosphorylation dependent, by creation of a phospho-degron.

Reviewer #3:This is an interesting paper that adds to our understanding of Wnt regulation in diverse contexts, though developmental and tumorigenesis settings are of particular interest here.There is already an abundance of negative regulation of Wnt signaling, with recent interest in the turnover of receptors by the Ubiqutin E3 ligases ZNRF3 and RNF43. In turn, these E3 ligases are turned over by R-spondin/lgr5 so that Wnt signaling is potentiated. In this current paper, an additional layer of regulation of Wnt receptors by a Transmembrane Phosphatase has been demonstrated. The PTPRK phosphatase removes phosphates from the ZNRF3 molecule, stimulating internalization, where it is proposed to carry the Fzd/LRP6 receptor complexes into the lysosome. Of interest to the tumor field, there are gene fusions between the adjacent PTPRK and RSPO3 that increase RSPO3 expression. Normally RSPO3 binds to Lgr5 and stimulates removal of ZNFR3/RNF43, thereby allowing the Wnt receptors to stay on the cell surface. The fusion also reduces the activity of PTPRK, so phosphorylated ZNRF3 no longer internalizes and destabilizes FZD/LRP6, also leading to increased Wnt signaling. Thus a new finding is that the gene fusion has two effects, both of which potentiate Wnt signaling, and promote tumor formation.Figure 1. The effect was discovered in an RNAi screen, and followed up carefully with Topflash, Axin2 expression, immunofluorescence, and Western blot tests for the function and abundance of β-catenin.Figure 2. shows the expression of the PTPRK gene in frog embryos, where it is enriched in the organizer and other tissues. Knockdown by splice blocking Mo or CRISPR causes a small head and truncated tail phenotype, typical of excess Wnt signaling, and TOPflash experiments confirm this.Figure 3 shows effects on organizer gene expression of the ptprk Mo, presumably because this interferes with the early cleavage Wnt signal that promotes organizer formation, in a secondary effect at the gastrula and neurula stage, the Mos cause restriction of anterior gene expression.Figure 4 shows that knocking down the E3 ligases or PTPRK the amount of Fzd or LRP6 on the cell surface.Figure 5 shows expression of znfr3 in coincident locations to ptprk in Xenopus embryos, and that both knockdowns work in a similar direction.Figure 6 shows biochemical interaction between ZNRF3 and PTPRK, and the consequences of knockdown of PTPRK on the subcellular distribution of ZNRF3, with more efficient removal of ZNFR3 to lysozomes in the presence of PTPRK.Supplementary figures show appropriate controls quantifying various effects.The supplemental summary figure is useful, though it might be a little clearer to show another (dephosphorylated) ZNRF3 molecule on its way to interact with the FZD LRP6 complex. The work is clearly presented, though with so many negative interactions, a little mind bending.

We have now modified our model for better understanding (Figure 7—figure supplement 2).